# The Sensory Neuron as a Transformer: Permutation-Invariant Neural Networks for Reinforcement Learning

**Yujin Tang**[†]
Google Brain
yujintang@google.com

**David Ha**[†]
Google Brain
hadavid@google.com

## Abstract

In complex systems, we often observe complex global behavior emerge from a collection of agents interacting with each other in their environment, with each individual agent acting only on locally available information, without knowing the full picture. Such systems have inspired development of artificial intelligence algorithms in areas such as swarm optimization and cellular automata. Motivated by the emergence of collective behavior from complex cellular systems, we build systems that feed each sensory input from the environment into distinct, but identical neural networks, each with no fixed relationship with one another. We show that these sensory networks can be trained to integrate information received locally, and through communication via an attention mechanism, can collectively produce a globally coherent policy. Moreover, the system can still perform its task even if the ordering of its inputs is randomly permuted several times during an episode. These permutation invariant systems also display useful robustness and generalization properties that are broadly applicable. Interactive demo and videos of our results: https://attentionneuron.github.io/

## 1 Introduction

Sensory substitution refers to the brain's ability to use one sensory modality (e.g., touch) to supply environmental information normally gathered by another sense (e.g., vision). Numerous studies have demonstrated that humans can adapt to changes in sensory inputs, even when they are fed into the *wrong* channels [5, 6, 25, 64]. But difficult adaptations–such as learning to "see" by interpreting visual information emitted from a grid of electrodes placed on one's tongue [6], or learning to ride a "backwards" bicycle [64]–require months of training to attain mastery. Can we do better, and create artificial systems that can rapidly adapt to sensory substitutions, without the need to be retrained?

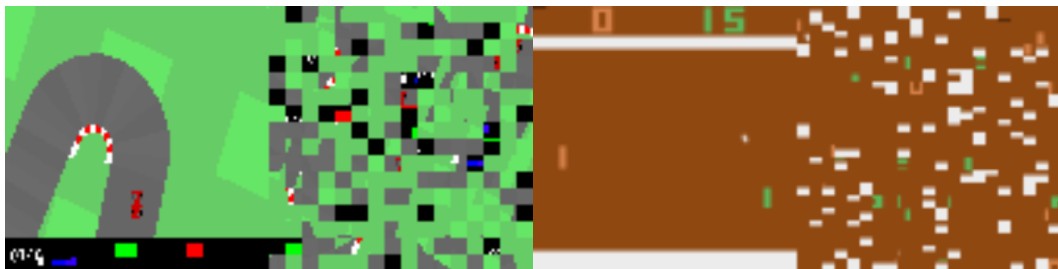

Figure 1: *Comparison of visual input intended for the game player, and what our system receives.* We partition the visual input from CarRacing (Left) and Atari Pong (right) into a 2D grid of small patches, and randomly permute their ordering. Each sensory neuron in the system receives a stream of visual input at a particular permuted patch location, and through coordination, must complete the task at hand, even if the visual ordering is randomly permuted again several times during an episode.

---

[†]Equal Contribution

35th Conference on Neural Information Processing Systems (NeurIPS 2021).

Modern deep learning systems are generally unable to adapt to a sudden reordering of sensory inputs, unless the model is retrained, or if the user manually corrects the ordering of the inputs for the model. However, techniques from continual meta-learning, such as adaptive weights [3, 36, 65], Hebbian-learning [52, 53, 57], and model-based [2, 20, 37, 38] approaches can help the model adapt to such changes, and remain a promising active area of research.

In this work, we investigate agents that are explicitly designed to deal with sudden random reordering of their sensory inputs while performing a task. Motivated by recent developments in self-organizing neural networks [27, 55, 61] related to cellular automata [14, 17, 18, 58, 78], in our experiments, we feed each sensory input (which could be an individual state from a continuous control environment, or a patch of pixels from a visual environment) into an individual neural network module that integrates information from only this particular sensory input channel over time. While receiving information locally, each of these individual sensory neural network modules also continually broadcasts an output message. Inspired by the Set Transformer [48, 74] architecture, an attention mechanism combines these messages to form a global latent code which is then converted into the agent's action space. The attention mechanism can be viewed as a form of adaptive weights of a neural network, and in this context, allows for an arbitrary number of sensory inputs that can be processed in any random order.

In our experiments, we find that each individual sensory neural network module, despite receiving only localized information, can still collectively produce a globally coherent policy, and that such a system can be trained to perform tasks in several popular reinforcement learning (RL) environments. Furthermore, our system can utilize a varying number of sensory input channels in any randomly permuted order, even when the order is shuffled again several times during an episode.

Permutation invariant systems have several advantages over traditional fixed-input systems. We find that encouraging a system to learn a coherent representation of a permutation invariant observation space leads to policies that are more robust and generalizes better to unseen situations. We show that, without additional training, our system continues to function even when we inject additional input channels containing noise or redundant information. In visual environments, we show that our system can be trained to perform a task even if it is given only a small fraction of randomly chosen patches from the screen, and at test time, if given more patches, the system can take advantage of the additional information to perform better. We also demonstrate that our system can generalize to visual environments with novel background images, despite training on a single fixed background. Lastly, to make training more practical, we propose a behavioral cloning scheme to convert policies trained with existing methods into a permutation invariant policy with desirable properties.

## 2   Related Work

**Self-organization** is a process where some form of global order emerges from local interactions between parts of an initially disordered system. It is also a property observed in cellular automata (CA) [17, 18, 58], which are mathematical systems consisting of a grid of cells that perform computation by having each cell communicate with its immediate neighbors and performing a local computation to update its internal state. Such local interactions are useful in modeling complex systems [78] and have been applied to model non-linear dynamics in various fields [14]. Cellular Neural Networks [16] were first introduced in the 1980s to use neural networks in place of the algorithmic cells in CA systems. They were applied to perform image processing operations with parallel computation. Eventually, the concept of self-organizing neural networks found its way into deep learning in the form of Graph Neural Networks (GNN) [62, 79].

Using modern deep learning tools, recent work demonstrates that *neural CA*, or self-organized neural networks performing only local computation, can generate (and re-generate) coherent images [55] and voxel scenes [70, 84], and even perform image classification [61]. Self-organizing neural network agents have been proposed in the RL domain [11, 12, 59, 60], with recent work demonstrating that shared local policies at the actuator level [43], through communicating with their immediate neighbors, can learn a global coherent policy for continuous control locomotion tasks. While existing CA-based approaches present a modular, self-organized solution, they are *not* inherently permutation invariant. In our work, we build on neural CA, and enable each cell to communicate beyond its immediate neighbors via an attention mechanism that enables permutation invariance.

**Meta-learning** recurrent neural networks (RNN) [23, 40, 42, 76] have been proposed to approach the problem of learning the learning rules for a neural network using the reward or error signal, enabling meta-learners to learn to solve problems presented outside of their original training domains. The

goals are to enable agents to continually learn from their environments in a single lifetime episode, and to obtain much better data efficiency than conventional learning methods such as stochastic gradient descent (SGD). A meta-learned policy that can adapt the weights of a neural network to its inputs during inference time have been proposed in fast weights [65, 67], associative weights [3], hypernetworks [36], and Hebbian-learning [52, 53] approaches. Recently works [46, 63] combine ideas of self-organization with meta-learning RNNs, and have demonstrated that modular meta-learning RNN systems not only can learn to perform SGD-like learning rules, but can also discover more general learning rules that transfer to classification tasks on unseen datasets.

In contrast, the system presented here does not use an error or reward signal to meta-learn or fine-tune its policy. But rather, by using the shared modular building blocks from the meta-learning literature, we focus on learning or converting an existing policy to one that is permutation invariant, and we examine the characteristics such policies exhibit in a zero-shot setting, *without* additional training.

**Attention** can be viewed as an adaptive weight mechanism that alters the weight connections of a neural network layer based on what the inputs are. Linear *dot-product* attention has first been proposed for meta-learning [66], and versions of linear attention with $softmax$ nonlinearity appeared later [33, 51], now made popular with Transformer [74]. The adaptive nature of attention provided the Transformer with a high degree of expressiveness, enabling it to learn inductive biases from large datasets and have been incorporated into state-of-the-art methods in natural language processing [9, 21], image recognition [22] and generation [26], audio and video domains [31, 44, 71].

Attention mechanisms have found many uses for RL [1, 13, 56, 68, 72, 83]. Our work here specifically uses attention to enable communication between arbitrary numbers of modules in an RL agent. While previous work [32, 45, 54, 75, 81, 85] utilized attention as a communication mechanism between independent neural network modules of a GNN, our work focuses on studying the permutation invariant properties of attention-based communication applied to RL agents. Related work [50] used permutation invariant critics to enhance performance of multi-agent RL. Building [34, 82], Set Transformers [48] investigated the use of attention explicitly for permutation invariant problems that deal with set-structured data, which have provided the theoretical foundation for our work.

## 3 Method

### 3.1 Background

Our goal is to devise an agent that is permutation invariant (PI) in the action space to the permutations in the input space. While it is possible to acquire a quasi-PI agent by training with randomly shuffled observations and hope the agent's policy network has enough capacity to memorize all the patterns, we aim for a design that achieves true PI even if the agent is trained with fix-ordered observations. Mathematically, we are looking for a non-trivial function $f(x) : \mathcal{R}^n \mapsto \mathcal{R}^m$ such that $f(x[s]) = f(x)$ for any $x \in \mathcal{R}^n$, and $s$ is any permutation of the indices $\{1, \cdots, n\}$. A different but closely related concept is permutation equivariance (PE) which can be described by a function $h(x) : \mathcal{R}^n \mapsto \mathcal{R}^n$ such that $h(x[s]) = h(x)[s]$. Unlike PI, the dimensions of the input and the output must equal in PE.

Self-attentions can be PE. In its simplest form, self-attention is described as $y = \sigma(QK^\top)V$ where $Q, K \in \mathcal{R}^{n \times d_q}, V \in \mathcal{R}^{n \times d_v}$ are the Query, Key and Value matrices and $\sigma(\cdot)$ is a non-linear function. In most scenarios, $Q, K, V$ are functions of the input $x \in \mathcal{R}^n$ (e.g. linear transformations), and permuting $x$ therefore is equivalent to permuting the rows in $Q, K, V$ and based on its definition it is straightforward to verify the PE property. Set Transformer [48] cleverly replaced $Q$ with a set of learnable seed vectors, so it is no longer a function of input $x$, thus enabling the output to become PI. A simple, intuitive explanation of the PI property of self-attention is available in Appendix A.1.

### 3.2 Sensory Neurons with Attention

To create PI agents, we propose to add an extra layer in front of the agent's policy network $\pi$, which accepts the current observation $o_t$ and the previous action $a_{t-1}$ as its inputs. We call this new layer AttentionNeuron, and Figure 2 gives an overview of our method. Inside AttentionNeuron, we model the observation $o_t$ as an arbitrarily ordered, variable-length list of sensory inputs, each of which is passed into its own *sensory neuron*, a neural network module. Each sensory neuron only has partial access to the agent's observation, at time $t$, the $i$th neuron can see only the $i$th component of the observation $o_t[i]$. Combined with the previous action $a_{t-1}$, each sensory neuron computes messages $f_k(o_t[i], a_{t-1})$ and $f_v(o_t[i])$ that are broadcast to the rest of the system. We then use attention to aggregate these messages into a *global latent code*, $m_t$, that is PI with respect to the inputs.

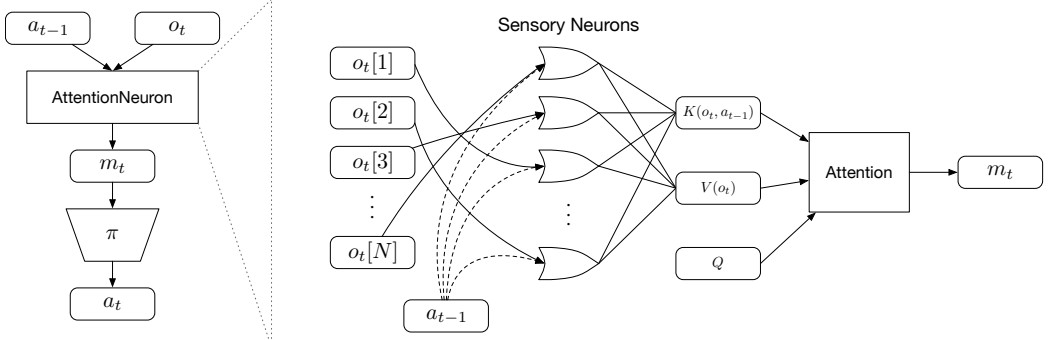

Figure 2: *Overview of Method.* AttentionNeuron is a standalone layer, in which each sensory neuron only has access to a part of the unordered observations $o_t$. Together with the agent's previous action $a_{t-1}$, each neuron generates messages independently using the shared functions $f_k(o_t[i], a_{t-1})$ and $f_v(o_t[i])$. The attention mechanism summarizes the messages into a global latent code $m_t$.

The operations inside AttentionNeuron can be described by the following two equations. For clarity, Table 1 summarizes the notations as well as the corresponding setups we used for the experiments.

$$K(o_t, a_{t-1}) = \begin{bmatrix} f_k(o_t[1], a_{t-1}) \\ \cdots \\ f_k(o_t[N], a_{t-1}) \end{bmatrix} \in \mathcal{R}^{N \times d_{f_k}}, V(o_t) = \begin{bmatrix} f_v(o_t[1]) \\ \cdots \\ f_v(o_t[N]) \end{bmatrix} \in \mathcal{R}^{N \times d_{f_v}} \quad (1)$$

$$m_t = \sigma\Big(\frac{[QW_q][K(o_t, a_{t-1})W_k]^\top}{\sqrt{d_q}}\Big)[V(o_t)W_v] \quad (2)$$

Equation 1 shows how each of the $N$ sensory neuron independently generates its messages $f_k$ and $f_v$, which are functions shared across all sensory neurons. Equation 2 shows the attention mechanism aggregate these messages. Note that although we could have absorbed the projection matrices $W_q, W_k, W_v$ into $Q, K, V$, we keep them in the equation to show explicitly the formulation. Equation 2 is almost identical to the simple definition of self-attention mentioned earlier. Following [48], we make our $Q$ matrix a bank of fixed embeddings, rather than depend on the observation $o_t$.

Note that permuting the observations only affects the row orders of $K$ and $V$, and that applying the same permutation to the rows of both $K$ and $V$ still results in the same $m_t$ which is PI. As long as we set constant the number of rows in $Q$, the change in the input size affects only the number of rows in $K$ and $V$ and does not affect the output $m_t$. In other words, our agent can accept inputs of arbitrary length and output a fixed sized $m_t$. Later, we apply this flexibility of input dimensions to RL agents.

Table 1: In this notation list, we provide the dimensions used in our model for different RL environments, to give the reader a sense of the relative magnitudes involved in each part of the system.

| Description | Notation | CartPole | Ant | CarRacing | Atari Pong |
|---|---|---|---|---|---|
| Full observation space | $o_t$ | $\mathcal{R}^5$ | $\mathcal{R}^{28}$ | $\mathcal{R}^{96 \times 96 \times 4}$ | $\mathcal{R}^{84 \times 84 \times 4}$ |
| Individual sensory input space | $o_t[i]$ | $\mathcal{R}^1$ | $\mathcal{R}^1$ | $\mathcal{R}^{6 \times 6 \times 4 = 144}$ | $\mathcal{R}^{6 \times 6 \times 4 = 144}$ |
| Number of sensory neurons | $N$ | 5 | 28 | $(96/6)^2 = 256$ | $(84/6)^2 = 196$ |
| Dimension of action space | $|A|$ | 1 | 8 | 3 | 6 (one-hot) |
| Number of embeddings in $Q$ | $M$ | 16 | 32 | 1024 | 400 |
| Projection matrix for Q | $W_q$ | $\mathcal{R}^{8 \times 32}$ | $\mathcal{R}^{8 \times 32}$ | $\mathcal{R}^{8 \times 16}$ | $\mathcal{R}^{8 \times 32}$ |
| Projection matrix for K | $W_k$ | $\mathcal{R}^{8 \times 32}$ | $\mathcal{R}^{8 \times 32}$ | $\mathcal{R}^{111 \times 16}$ | $\mathcal{R}^{114 \times 32}$ |
| Projection matrix for V | $W_v$ | $I$ | $I$ | $\mathcal{R}^{144 \times 16}$ | $\mathcal{R}^{144 \times 32}$ |
| Post-attention activation function | $\sigma(\cdot)$ | $tanh$ | $tanh$ | $softmax$ | $softmax$ |
| Global latent code | $m_t$ | $\mathcal{R}^{16}$ | $\mathcal{R}^{32}$ | $\mathcal{R}^{1024 \times 16}$ | $\mathcal{R}^{400 \times 32}$ |

### 3.3 Design Choices

It is worthwhile to have a discussion on the design choices made. Since the ordering of the input is arbitrary, each sensory neuron is required to interpret and identify their received signal. To achieve this, we want $f_k(o_t[i], a_{t-1})$ to have temporal memories. In practice, we find both RNNs and feed-forward neural networks (FNN) with stacked observations work well, with FNNs being more practical for environments with high dimensional observations.

In addition to the temporal memory, including previous actions is important for the input identification too. Although the former allows the neurons to infer the input signals based on the characteristics of the temporal stream, this may not be sufficient. For example, when controlling a legged robot, most of the sensor readings are joint angles and velocities from the legs, which are not only numerically identically bounded but also change in similar patterns. The inclusion of previous actions gives each sensory neuron a chance to infer the casual relationship between the input channel and the applied actions, which helps with the input identification.

Finally, in Equation 2 we could have combined $QW_q \in \mathcal{R}^{M \times d_q}$ as a single learnable parameters matrix, but we separate them for two reasons. First, by factoring into two matrices, we can reduce the number of learnable parameters. Second, we find that instead of making $Q$ learnable, using the positional encoding proposed in Transformer [74] encourages the attention mechanism to generate distinct codes. Here we use the row indices in $Q$ as the positions for encoding.

## 4 Experiments

We experiment on several different RL environments to study various properties of permutation invariant RL agents. Due to the nature of the underlying tasks, we will describe the different architectures of the policy networks used and discuss various different training methods. However, the AttentionNeuron layers in all agents are similar, so we first describe the common setups. Hyper-parameters and other details for all experiments are summarized in Appendix A.4.

For non-vision continuous control tasks, the agent receives an observation vector $o_t \in \mathcal{R}^{|O|}$ at time $t$. We assign $N = |O|$ sensory neurons for the tasks, each of which sees one element from the vector, hence $o_t[i] \in \mathcal{R}^1, i = 1, \cdots, |O|$. We use an LSTM [41] as our $f_k(o_t[i], a_{t-1})$ to generate Keys, the input size of which is $1 + |A|$ (2 for Cart-Pole and 9 for PyBullet Ant). A simple pass-through function $f(x) = x$ serves as our $f_v(o_t[i])$, and $\sigma(\cdot)$ is $tanh$. For simplicity, we find $W_v = I$ works well for the tasks, so the learnable components are the LSTM, $W_q$ and $W_k$.

For vision based tasks, we gray-scale and stack $k = 4$ consecutive RGB frames from the environment, and thus our agent observes $o_t \in \mathcal{R}^{H \times W \times k}$. $o_t$ is split into non-overlapping patches of size $P = 6$ using a sliding window, so each sensory neuron observes $o_t[i] \in \mathcal{R}^{6 \times 6 \times k}$. Here, $f_v(o_t[i])$ flattens the data and returns it, hence $V(o_t)$ returns a tensor of shape $N \times d_{f_v} = N \times (6 \times 6 \times 4) = N \times 144$. Due to the high dimensionality for vision tasks, we do not use RNNs for $f_k$, but instead use a simpler method to process each sensory input. $f_k(o_t[i], a_{t-1})$ takes the difference between consecutive frames $(o_t[i])$, then flattens the result, appends $a_{t-1}$, and returns the concatenated vector. $K(o_t, a_{t-1})$ thus gives a tensor of shape $N \times d_{f_k} = N \times [(6 \times 6 \times 3) + |A|] = N \times (108 + |A|)$ (111 for CarRacing and 114 for Atari Pong). We use $softmax$ as the non-linear activation function $\sigma(\cdot)$, and we apply layer normalization [4] to both the input patches and the output latent code.

### 4.1 Cart-pole swing up

We examine Cart-pole swing up [29, 30, 35, 86] to first illustrate our method, and also use it to provide a clear analysis of the attention mechanism. We use `CartPoleSwingUpHarder` [28], a more difficult version of the task where the initial positions and velocities are highly randomized, leading to a higher variance of task scenarios. In the environment, the agent observes $[x, \dot{x}, cos(\theta), sin(\theta), \dot{\theta}]$, outputs a scalar action, and is rewarded at each step for getting $x$ close to 0 and $cos(\theta)$ close to 1.

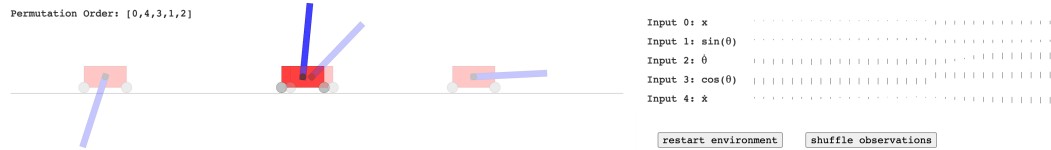

Figure 3: *Interactive demo of CartPoleSwingUpHarder.* In our web demo[2], the user can shuffle the order of the 5 inputs at any time, and observe how the agent adapts to the new ordering of the inputs.

We use a two-layer neural network as our agent. The first layer is an AttentionNeuron layer with $N = 5$ sensory neurons and outputs $m_t \in \mathcal{R}^{16}$. A linear layer takes $m_t$ as input and outputs a scalar action. For comparison, we also trained an agent with a two-layer FNN policy with 16 hidden units. We use direct policy search to train agents with CMA-ES [39], an evolution strategies (ES) method.

---

[2]Interactive demos and videos of our results available at `https://attentionneuron.github.io/`

Table 2: *Cart-pole Tests.* For each experiment, we report the average score and the standard deviation from 1000 test episodes. Our agent is trained only in the environment with 5 sensory inputs.

|  | 5 obs | 5 obs (shuffled) | 10 obs | 5 obs + 5 noise |
|---|---|---|---|---|
| FNN (trained with 5 obs) | $593 \pm 433$ | $38 \pm 120$ | N/A | N/A |
| FNN (trained with 10 obs) | N/A | N/A | $593 \pm 433$ | $137 \pm 242$ |
| Ours (trained with 5 obs) | $472 \pm 426$ | $471 \pm 426$ | $471 \pm 425$ | $461 \pm 410$ |

Our agent can perform the task and balance the cart-pole from an initially random state. Its average score is slightly lower than the baseline (See column 1 of Table 2) because each sensory neuron requires some time steps in each episode to interpret the sensory input signal it receives. However, as a trade-off for the performance sacrifice, our agent can retain its performance even when the input sensor array is randomly shuffled, which is not the case for an FNN policy (column 2). Moreover, although our agent is only trained in an environment with five inputs, it can accept an arbitrary number of inputs in any order without re-training[3]. We test our agent by duplicating the 5 inputs to give the agent 10 observations (column 3). When we replace the 5 extra signals with white noises with $\sigma = 0.1$ (column 4), we do not see a significant drop in performance.

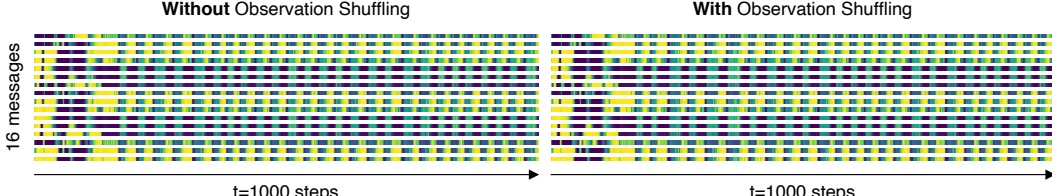

Figure 4: *Permutation invariant outputs.* The output (16-dimensional global latent code) from the AttentionNeuron layer does not change when we input the sensor array as-is (left) or when we randomly shuffle the array (right). Yellow represents higher values, and blue for lower values.

The AttentionNeuron layer should possess 2 properties to attain these: its output is permutation invariant to its input, and its output carries task-relevant information. Figure 4 is a visual confirmation of the permutation invariant property, whereby we plot the output messages from the layer and their changes over time from two tests. Using the same random seed, we keep the observation as-is in the first test but we shuffle the order in the second. As the figure shows, the output messages are identical in the two roll-outs. We also perform a simple linear regression analysis on the outputs (based on the shuffled inputs) to recover the 5 inputs in their original order. Table 3 shows the $R^2$ values[4] from this analysis, suggesting that some important indicators (e.g. $\dot{x}$ and $\dot{\theta}$) are well represented in the output.

Table 3: *Linear regression analysis on the output.* For each of the $N = 5$ sensory inputs we have one linear regression model with $m_t \in \mathcal{R}^{16}$ as the explanatory variables.

|  | $x$ | $\dot{x}$ | $cos(\theta)$ | $sin(\theta)$ | $\dot{\theta}$ |
|---|---|---|---|---|---|
| $R^2$ | 0.354 | 0.620 | 0.626 | 0.233 | 0.550 |

Table 4: *PyBullet Ant results.*

|  | Score |
|---|---|
| FNN (teacher) | $2700 \pm 28$ |
| FNN (shuffled) | $232 \pm 112$ |
| Ours (ES, shuffled) | $2576 \pm 75$ |
| Ours (BC, shuffled) | $2034 \pm 948$ |
| Ours (BC, shuffled, larger) | $2579 \pm 457$ |

## 4.2 PyBullet Ant

While direct policy search methods such as evolution strategies (ES) can train permutation invariant RL agents, oftentimes we already have access to pre-trained agents or recorded human data performing the task at hand. Behavior cloning (BC) can allow us to convert an existing policy to a version that is permutation invariant with desirable properties associated with it.

In Table 4, we train a standard two-layer FNN policy to perform `AntBulletEnv-v0`, a 3D locomotion task in PyBullet [19], and use it as a teacher for BC. For comparison, we also train a two-layer agent with AttentionNeuron for its first layer. Both networks are trained with ES. Similar to CartPole, we expect to see a small performance drop due to some time steps required for the agent to interpret an arbitrarily ordered observation space. We then collect data from the FNN teacher policy to train permutation invariant agents using BC. More details of the BC setup can be found in Appendix A.4.2.

---

[3]Because our agent was not trained with normalization layers, we scaled the output from the AttentionNeuron layer by 0.5 to account for the extra inputs in the last 2 experiments.

[4]$R^2$ measures the goodness-of-fit of a model. An $R^2$ of 1 implies that the regression perfectly fits the data.

The performance of the BC agent is lower than the one trained from scratch with ES, despite having the identical architecture. This suggests that the inductive bias that comes with permutation invariance may not match the original teacher network, so the small model used here may not be expressive enough to clone any teacher policy, resulting in a larger variance in performance. A benefit of gradient-based BC, compared to RL, is that we can easily train larger networks to fit the behavioral data. We show that increasing the size of the subsequent layers for BC does enhance the performance.

As we will demonstrate next, BC is a useful technique for training permutation invariant agents in environments with high dimensional visual observations that may require larger networks.

### 4.3 Atari Pong

Here, we are interested in solving screen-shuffled versions of vision-based RL environments, where each observation frame is divided up into a grid of patches, and like a puzzle, the agent must process the patches in a shuffled order to determine a course of action to take. A shuffled version of Atari Pong [8] (See Figure 1, right pair) can be especially hard for humans to play when inductive biases from human priors [24] that expect a certain type of spatial structure is missing from the observations.

But rather than throwing away the spatial structure entirely from our solution, we find that convolution neural network (CNN) policies work better than fully connected multi-layer perceptron (MLP) policies when trained with behavior cloning for Atari Pong. In this experiment, we reshape the output $m_t$ of the AttentionNeuron layer from $\mathcal{R}^{400 \times 32}$ to $\mathcal{R}^{20 \times 20 \times 32}$, a 2D grid of latent codes, and pass this 2D grid into a CNN policy. This way, the role of the AttentionNeuron layer is to take a list of unordered observation patches, and learn to construct a 2D grid representation of the inputs to be used by a downstream policy that expects some form of spatial structure in the codes. Our permutation invariant policy trained with BC can consistently reach a perfect score of 21, even with shuffled screens. The details of the CNN policy and BC training can be found in Appendix A.4.3.

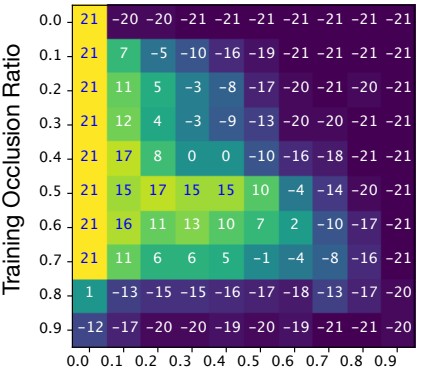
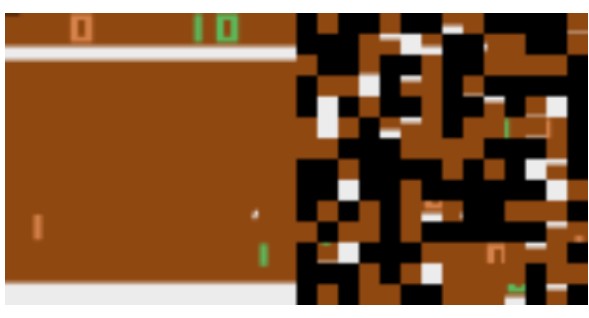

Figure 5: *Mean test scores in Atari Pong, and example of a randomly-shuffled occluded observation.* In the heat map, each value is the average score from 100 test episodes. For comparison, we show the original screen (left), and the agent's observation (right). Discarded patches shown here in black.

Unlike typical CNN policies, our agent can accept a subset of the screen, since the agent's input is a variable-length list of patches. It would thus be interesting to deliberately randomly discard a certain percentage of the patches and see how the agent reacts. The net effect of this experiment for humans is similar to being asked to play a partially occluded and shuffled version of Atari Pong (see Figure 5, right). During training via BC, we randomly remove a percentage of observation patches. In tests, we fix the randomly selected positions of patches to discard during an entire episode.

We present the results in a heat map in Figure 5 (left), where the y-axis shows the patches removed during training and the x-axis gives the patch occlusion ratio in tests. The diagram shows clear patterns for interpretation. Looking horizontally along each row, the performance drops because the agent sees less of the screen which increases the difficulty. Interestingly, an agent trained at a high occlusion rate of $80\%$ rarely wins against the Atari opponent, but once it is presented with the full set of patches during tests, it is able to achieve a fair result by making use of the additional information.

To gain insights into understanding the policy, we projected the AttentionNeuron layer's output in a test roll-out to 2D space using t-SNE [73]. In Figure 6, we highlight several groups and show their

corresponding inputs. The AttentionNeuron layer clearly learned to cluster inputs that share similar features. For example, the 3 sampled inputs in the blue group show the situation when the agent's paddle moved toward the bottom of the screen and stayed there. Similarly, the orange group shows the cases when the ball was not in sight, this happened right before/after a game started/ended. We believe these discriminative outputs enabled the downstream policy to accomplish the agent's task.

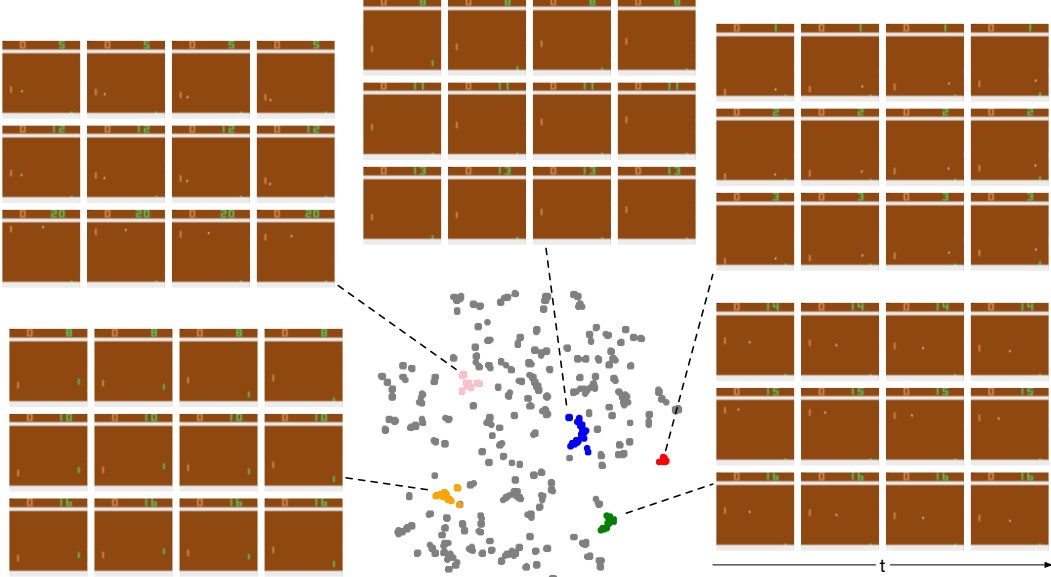

Figure 6: *2D embedding of the AttentionNeuron layer's output in a test episode.* We highlight several representative groups in the plot, and show the sampled inputs from them. For each group, we show 3 corresponding inputs (rows) and unstack each to show the time dimension (columns).

## 4.4 CarRacing

We find that encouraging an agent to learn a coherent representation of a deliberately shuffled visual scene leads to agents with useful generalization properties. Such agents are still able to perform their task even if the visual background of the environment changes, despite being trained only on a single static background. Out-of-domain generalization is an active area, and here, we combine our method with AttentionAgent [72], a method that uses selective, hard-attention via a patch voting mechanism. AttentionAgents in [72] generalize well to several unseen visual environments where task irrelevant elements are modified, but fail to generalize to drastic background changes in a zero-shot setting.

In this experiment, we combine the permutation invariant AttentionNeuron layer with the policy network used in AttentionAgent. As their hard-attention-based policy is non-differentiable, we train the entire system using ES. We reshape the AttentionNeuron layer's outputs to adapt for the policy network. Specifically, we reshape the output message to $m_t \in \mathcal{R}^{32 \times 32 \times 16}$ such that it can be viewed as a 32-by-32 grid of 16 channels. The end result is a policy with two layers of attention: the first layer outputs a latent code book to represent a shuffled scene, and the second layer performs hard attention to select the top $K = 10$ codes from a 2D global latent code book. A detailed description of the selective hard attention policy from [72] and other training details can be found in Appendix A.4.4.

We first train the agent in the CarRacing [47] environment, and report the average score from 100 test roll-outs in Table 5. As the first column shows, our agent's performance in the training environment is slightly lower but comparable to the baseline method, as expected. But because our agent accepts randomly shuffled inputs, it is still able to navigate even when the patches are shuffled. Figure 1 (left pair) gives an illustration, where the right screen is what our agent observes and the left is for human visualization. A human will find driving with the shuffled observation to be very difficult because we are not constantly exposed to such tasks, just like in the "reverse bicycle" example mentioned earlier.

Without additional training or fine-tuning, we test whether the agent can also navigate in four modified environments where the green grass background is replaced with various images (See Figure 7). As Table 5 (from column 2) shows, our agent generalizes well to most of the test environments with only mild performance drops while the baseline method fails to generalize. We suspect this is because

Table 5: CarRacing Test Results.

| | Training Env | KOF | Mt. Fuji | Ukiyoe | DS |
|---|---|---|---|---|---|
| AttentionAgent [72] | $901 \pm 54$ | $-81 \pm 4$ | $-57 \pm 38$ | $-107 \pm 50$ | $-56 \pm 23$ |
| NetRand [49] | $480 \pm 144$ | $20 \pm 84$ | $356 \pm 159$ | $533 \pm 111$ | $-27 \pm 34$ |
| NetRand + AttentionAgent | $885 \pm 64$ | $-51 \pm 14$ | $709 \pm 94$ | $656 \pm 131$ | $122 \pm 134$ |
| Ours | $801 \pm 147$ | $646 \pm 189$ | $503 \pm 152$ | $661 \pm 140$ | $171 \pm 146$ |

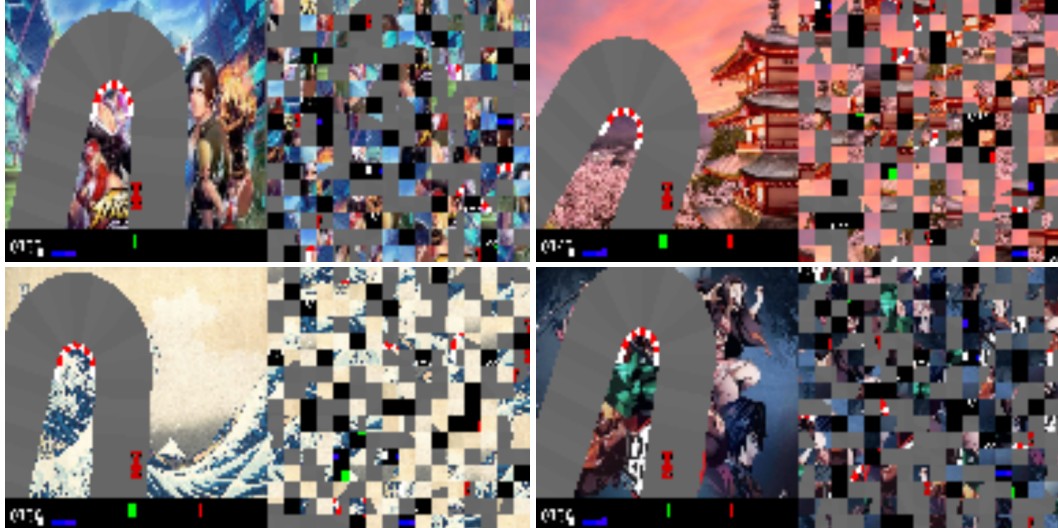

Figure 7: *Screenshots of test environments.* In each pair of images, the left is for human visualization and the right is what our agent sees. From the top left and in the clockwise order, the environments are "KOF", "Mt. Fuji", "DS" and "Ukiyoe".

the AttentionNeuron layer has transformed the original RGB space to a useful hidden representation (represented by $m_t$) that has eliminated task irrelevant information after observing and reasoning about the sequences of $(o_t, a_{t-1})$ during training, enabling the downstream hard attention policy to work with an optimized abstract representation tailored for the policy, instead of raw RGB patches.

We also compare our method to NetRand [49], a simple but effective technique developed to perform similar generalization tasks. In the second row of Table 5 are the results of training NetRand on the base CarRacing task. The CarRacing task proved to be too difficult for NetRand, but despite a low performance score of 480 in the training environment, the agent generalizes well to the "Mt. Fuji" and "Ukiyoe" modifications. In order to obtain a meaningful comparison, we combine NetRand with AttentionAgent so that it can get close to a mean score of 900 on the base task. To do that, we use NetRand as an input layer to the AttentionAgent policy network, and train the combination end-to-end using ES, which is consistent with our proposed method for this task. The combination attains a respectable mean score of 885, and as we can see in the third row of the above table, this approach also generalizes to a few of the unseen modifications of the CarRacing environment.

Our score on the base CarRacing task is lower than NetRand, but this is expected since our agent requires some amount of time steps to identify each of the inputs (which could be shuffled), while the NetRand and AttentionAgent agent will simply fail on the shuffled versions of CarRacing. Despite this, our method still compares favorably on the generalization performance.

We visualize the attentions from the AttentionNeuron layer in Figure 8. In CarRacing, the agent has learned to focus its attention (indicated by the highlighted patches) on the road boundaries which are intuitive to human beings and are critical to the task. Notice that the attended positions are consistent before and after the shuffling. More details about this visualization can be found in Appendix A.4.4.

## 5 Discussion and Future Work

In this work, we investigate the properties of RL agents that can treat their observations as an arbitrarily ordered, variable-length list of sensory inputs. By processing each input stream independently, and consolidating the processed information using attention, our agents can still perform their tasks even if the ordering of the observations is randomly permuted several times during an episode, without explicitly training for frequent re-shuffling (See Table 6).

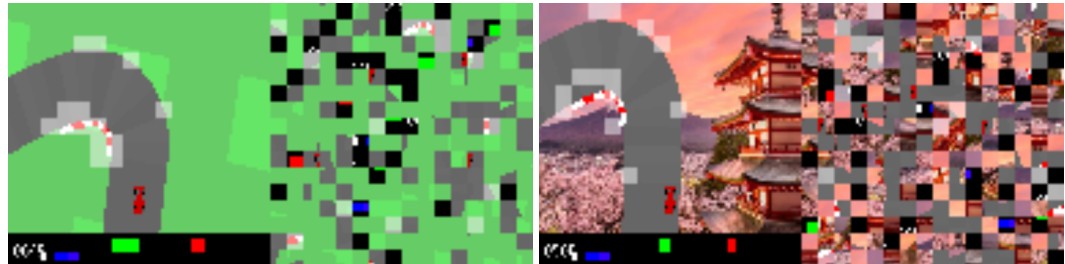

Figure 8: *Attention visualization.* We highlight the observed patches that receive the most attention. Left: Training environment. Right: Test environment with unseen background.

Table 6: *Reshuffle observations during a roll-out.* In each test episode, we reshuffle the observations every $t$ steps. For CartPole, we test for 1000 episodes because of its larger task variance. For the other tasks, we report mean and standard deviation from 100 tests. All environments except for Atari Pong have a hard limit of 1000 time steps per episode. In Atari Pong, while the maximum length of an episode does not exist, we observed that an episode usually lasts for around 2500 time steps.

|  | CartPole | PyBullet Ant | Atari Pong | CarRacing |
|---|---|---|---|---|
| $t = 25$ | $107 \pm 146$ | $2053 \pm 225$ | $-20 \pm 1$ | $732 \pm 161$ |
| $t = 50$ | $163 \pm 198$ | $2319 \pm 188$ | $-20 \pm 2$ | $772 \pm 163$ |
| $t = 100$ | $242 \pm 254$ | $2406 \pm 178$ | $-10 \pm 12$ | $768 \pm 167$ |
| $t = 200$ | $318 \pm 310$ | $2493 \pm 105$ | $-2 \pm 17$ | $774 \pm 182$ |
| $t = 500$ | $407 \pm 380$ | $2548 \pm 87$ | $18 \pm 9$ | $805 \pm 158$ |
| No reshuffle | $472 \pm 426$ | $2576 \pm 75$ | $21 \pm 0$ | $801 \pm 147$ |

**Applications**  By presenting the agent with shuffled, and even incomplete observations, we encourage it to interpret the meaning of each local sensory input and how they relate to the global context. This could be useful in many real world applications. For example, such policies could avoid errors due to cross-wiring or complex, dynamic input-output mappings when being deployed in real robots. A similar setup to the CartPole experiment with extra noisy channels could enable a system that receives thousands of noisy input channels to identify the small subset of channels with relevant information.

**Limitations**  For visual environments, patch size selection will affect both performance and computing complexity. We find that patches of 6x6 pixels work well for our tasks, as did 4x4 pixels to some extent, but single pixel observations fail to work. Small patch sizes also result in a large attention matrix which may be too costly to compute, unless approximations are used [15, 77, 80].

Another limitation is that the permutation invariant property applies only to the inputs, and not to the outputs. While the ordering of the observations can be shuffled, the ordering of the actions cannot. For permutation invariant outputs to work, each action will require feedback from the environment, including reward information, in order to learn the relationship between itself and the environment.

**Societal Impact**  Like most algorithms proposed in computer science and machine learning, our method can be applied in ways that will have potentially positive or negative impacts to society. While our small-scale, self-contained experiments study only the properties of agents that are PI to their observations, and we believe our results do not directly cause harm to society, the robustness and flexible properties of the method may be of use for data-collection systems that receive data from a large variable number of sensors. For instance, one could apply permutation invariant sensory systems to process data from millions of sensors for anomaly detection, which may lead to both positive or negative impacts, if used in applications such as large-scale sensor analysis for weather forecasting, or deployed in large-scale surveillance systems that could undermine our basic freedoms.

Our work also provides a way to view the Transformer [74] through the lens of self-organizing neural networks. Transformers are known to have potentially negative societal impacts highlighted in studies about possible data-leakage and privacy vulnerabilities [10], malicious misuse and issues concerning bias and fairness [7], and energy requirements for training them [69].

**Future Work**  An interesting future direction is to also make the action layer have the same properties, and model each "motor neuron" as a module connected using attention. With such methods, it may be possible to train an agent with an arbitrary number of legs, or control robots with different morphology using a single policy that is also provided with a reward signal as feedback. We look forward to seeing future works that include signals such as environmental rewards to train PI meta-learning agents that can adapt to not only changes in the observed environment, but also to changes to itself.

## Acknowledgements

The authors would like to thank Rishabh Agarwal, Jie Tan, Yingtao Tian, Douglas Eck, Aleksandra Faust and our NeurIPS2021 reviewers for valuable discussion and feedback. The experiments in this work were conducted using virtual machines provided by Google Cloud Platform.

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
