# A Appendix

## A.1 Intuitive explanation of Self-Attention's permutation invariant property

Here, we provide a simple, non-rigorous example demonstrating permutation invariant property of the self-attention mechanism, to give some intuition to readers who may not be familiar with self-attention. For a detailed treatment, please refer to [1].

As mentioned in Section 3.1 of the main text, in its simplest form, self-attention is described as:

$$y = \sigma(QK^\top)V \tag{1}$$

where $Q \in \mathcal{R}^{N_q \times d_q}, K \in \mathcal{R}^{N \times d_q}, V \in \mathcal{R}^{N \times d_v}$ are the Query, Key and Value matrices and $\sigma(\cdot)$ is a non-linear function. In this work, $Q$ is a fixed matrix, and $K, V$ are functions of the input $X \in \mathcal{R}^{N \times d_{in}}$ where $N$ is the number of observation components (equivalent to the number of sensory neurons) and $d_{in}$ is the dimension of each component. In most settings, $K = XW_k, V = XW_v$ are linear transformations, thus permuting $X$ therefore is equivalent to permuting the rows in $K, V$.

We would like to show that the output $y$ is the same regardless of the ordering of the rows of $K, V$. For simplicity, suppose $N = 3, N_q = 2, d_q = d_v = 1$, so that $Q \in \mathcal{R}^{2 \times 1}, K \in \mathcal{R}^{3 \times 1}, V \in \mathcal{R}^{3 \times 1}$:

$$
\begin{aligned}
y &= \sigma\left( \begin{pmatrix} q_1 \\ q_2 \end{pmatrix} (k_1 \; k_2 \; k_3) \right) \begin{pmatrix} v_1 \\ v_2 \\ v_3 \end{pmatrix} \\
&= \sigma\left( \begin{pmatrix} q_1k_1 & q_1k_2 & q_1k_3 \\ q_2k_1 & q_2k_2 & q_2k_3 \end{pmatrix} \right) \begin{pmatrix} v_1 \\ v_2 \\ v_3 \end{pmatrix} \\
&= \begin{pmatrix} \sigma(q_1k_1)v_1 \; + \; \sigma(q_1k_2)v_2 \; + \; \sigma(q_1k_3)v_3 \\ \sigma(q_2k_1)v_1 \; + \; \sigma(q_2k_2)v_2 \; + \; \sigma(q_2k_3)v_3 \end{pmatrix}
\end{aligned} \tag{2}
$$

The output $y \in \mathcal{R}^{2 \times 1}$ remains the same when the rows of $K, V$ are permuted from $[1, 2, 3]$ to $[3, 1, 2]$:

$$
\begin{aligned}
y &= \sigma\left( \begin{pmatrix} q_1 \\ q_2 \end{pmatrix} (k_3 \; k_1 \; k_2) \right) \begin{pmatrix} v_3 \\ v_1 \\ v_2 \end{pmatrix} \\
&= \sigma\left( \begin{pmatrix} q_1k_3 & q_1k_1 & q_1k_2 \\ q_2k_3 & q_2k_1 & q_2k_2 \end{pmatrix} \right) \begin{pmatrix} v_3 \\ v_1 \\ v_2 \end{pmatrix} \\
&= \begin{pmatrix} \sigma(q_1k_3)v_3 \; + \; \sigma(q_1k_1)v_1 \; + \; \sigma(q_1k_2)v_2 \\ \sigma(q_2k_3)v_3 \; + \; \sigma(q_2k_1)v_1 \; + \; \sigma(q_2k_2)v_2 \end{pmatrix}
\end{aligned} \tag{3}
$$

We have highlighted the same terms with the same color in Equations 2 and 3 to show the results are indeed identical. In general, we have $y_{ij} = \sum_{b=1}^{N} \sigma\left(\sum_{a=1}^{d_q} Q_{ia}K_{ba}\right)V_{bj}$. Permuting the input is equivalent to permuting the indices $b$ (i.e. rows of $K$ and $V$), which only affects the order of the outer summation and does not affect $y_{ij}$ because summation is a permutation invariant operation. Notice that in the above example and the proof here we have assumed that $\sigma(\cdot)$ is an element-wise operation—a valid assumption since most activation functions satisfy this condition.[1]

As discussed in Section 3.2 of the main text, this formulation lets us convert an observation signal from the RL environment into a permutation invariant representation $y$. We can use $y$ in place of the actual observation as the input that goes into the downstream policy network of an RL agent.

## A.2 Hyper-parameters

Table 1 in the main text contains the hyper-parameters used for each experiment. We did not employ exhaustive hyper-parameter tuning, but have simply selected (from experience) hyper-parameters that work with training methods such as evolution strategies, where the number of model parameters cannot be too large. As mentioned in the discussion section about the limitations, we tested a small range of patch sizes (1, 4, and 6 pixels), and we find that a patch size of 6x6 works well across tasks.

---

[1]Applying *softmax* to each row only brings scalar multipliers to each row and the proof still holds.

### A.3 Description of compute infrastructure used to conduct experiments

For all ES results, we train on Google Kubernetes Engines (GKE) with 256 CPUs (N1 series) for each job. The approximate time, including both training and periodic tests, for the jobs are: 3 days (CartPole), 5 days (PyBullet Ant ES) and 10 days (CarRacing). For BC results, we train with Google Computing Engines (GCE) on an instance that has one V100 GPU. The approximate time, including both training and periodic tests, for the jobs are: 5 days (PyBullet Ant BC), 1 day (Atari Pong).

### A.4 Detailed setups for the experiments

#### A.4.1 Training budget

The costs of ES training are summarized in the following table. A maximum of 20K generations is specified in the training, but stopped early if the performance converged. Each generation has $256 \times 16 = 4096$ episode rollouts, where 256 is the population size and 16 is the rollout repetitions. The Pong permutation-invariant (PI) agents were trained using behavior cloning (BC) on a pre-trained PPO policy (which is not PI-capable), with 10M training steps.

| Environment | CartPoleSwingUpHarder | PyBullet Ant | Atari Pong | CarRacing |
|---|---|---|---|---|
| Number of Generations | 14,000 | 12,000 | - | 4,000 |

Note that we used the hyper-parameters (e.g., population size, rollout repetitions) that proved to work on a wide range of tasks from past experience, and did not tune them for each experiment. In other words, these settings were not chosen with sample-efficiency in mind, but rather for learning a working PI-capable policy using distributed computation within a reasonable wall-clock time budget.

We consider two possible approaches when we take sample-efficiency into consideration. In the experiments, we have demonstrated that it is possible to simply use state-of-the-art RL algorithms to learn a non-PI policy, and then use BC to produce a PI version of the policy. The first approach is thus to rely on the conventional RL algorithms to increase sample efficiency, which is a hot and on-going topic in the area. On the other hand, we do think that an interesting future direction is to formulate environments where BC will fail in a PI setting, and that interactions with the environment (in a PI setting) is required to learn a PI policy. For instance, we have demonstrated in PyBullet Ant that the BC method requires the cloned agent to have a much larger number of parameters compared to one trained with RL. This is where an investigation in sample-efficiency improvements in the RL algorithm explicitly in the PI setting may be beneficial.

#### A.4.2 PyBullet Ant

In the PyBullet Ant experiment, we demonstrated that a pre-trained policy can be converted into a permutation invariant one with behavior cloning (BC). We give detailed task description and experimental setups here. In `AntBulletEnv-v0`, the agent controls an ant robot that has 8 joints ($|A| = 8$), and gets to see an observation vector that has base and joint states as well as foot-ground contact information at each time step (|O|=28). The mission is to make the ant move along a pre-defined straight line as fast as possible. The teacher policy is a 2-layer FNN policy that has 32 hidden units trained with ES. We collected data from 1000 test roll-outs, each of which lasted for 500 steps. During training, we add zero-mean Gaussian noise ($\sigma = 0.03$) to the previous actions. For the student policy, We set up two networks. The first policy is a 2-layered network that has the AttentionNeuron with output size $m_t \in \mathcal{R}^{32}$ as its first layer, followed by a fully-connected (FC) layer. The second, larger policy is similar in architecture, but we added one more FC layer and expanded all hidden size to 128 to increase its expressiveness. We train the students with a batch size of 64, an Adam optimizer of $lr = 0.001$ and we clip the gradient at maximum norm of 0.5.

#### A.4.3 Atari Pong

In the Atari game Pong, we append a deep CNN to the AttentionNeuron layer in our agent (student policy). To be concrete, we reshape the AttentionNeuron's output message $m_t \in \mathcal{R}^{400 \times 32}$ to $m_t \in \mathcal{R}^{20 \times 20 \times 32}$ and pass it to the trailing CNN: [Conv(in=32, out=64, kernel=4, stride=2), Conv(in=64, out=64, kernel=3, stride=1), FC(in=3136, out=512), FC(in=512, out=6)]. We use $ReLU$ as the activation functions in the CNN. We collect the stacked observations and the corresponding logits

output from a pre-trained PPO agent (teacher policy) from 1000 roll-outs, and we minimize the MSE loss between the student policy's output and the teacher policy's logits. The learning rate and norm clip are the same as the previous experiment, but we use a batch size of 256.

For the occluded Pong experiment, we randomly remove a certain percentage of the patches across a training batch of stacked observation patches. In tests, we sample a patch mask to determine the positions to occlude at the beginning of the episode, and apply this mask throughout the episode.

### A.4.4   CarRacing

In AttentionAgent [2], the authors observed that the agent generalizes well if it is forced to make decisions based on only a fraction of the available observations. Concretely, [2] proposed to segment the input image into patches and let the patches vote for each other via a modified self-attention mechanism. The agent would then take into consideration only the top $K = 10$ patches that have the most votes and based on the coordinates of which an LSTM controller makes decisions. Because the voting process involves sorting and pruning that are not differentiable, the agent is trained with ES. In their experiments, the authors demonstrated that the agent could navigate well not only in the training environment, but also zero-shot transfer to several modified environments.

We need only to reshape the AttentionNeuron layer's outputs to adapt for AttentionAgent's policy network. Specifically, we reshape the output message $m_t \in \mathcal{R}^{1024 \times 16}$ to $m_t \in \mathcal{R}^{32 \times 32 \times 16}$ such that it can be viewed as a 32-by-32 "image" of 16 channels. Then if we make AttentionAgent's patch segmentation size 1, the original patch voting becomes voting among the $m_t$'s and thus the output fits perfectly into the policy network. Except for this patch size, we kept all hyper-parameters in AttentionAgent unchanged, we also used the same CMA-ES training hyper-parameters.

Although the simple settings above allows our augmented agent to learn to drive and generalize to unseen background changes, we found the car jittered left and right through the courses. We suspect this is because of the frame differential operation in our $f_k(o_t, a_{t-1})$. Specifically, even when the car is on a straight lane, constantly steering left and right allows $f_k(o_t, a_{t-1})$ to capture more meaningful signals related to the changes of the road. To avoid such jittering behavior, we make $m_t$ a rolling average of itself: $m_t = (1 - \alpha)m_t + \alpha m_{t-1}, 0 \leq \alpha \leq 1$. In our implementation $\alpha = g([h_{t-1}, a_{t-1}])$, where $h_{t-1}$ is the hidden state from AttentionAgent's LSTM controller and $a_{t-1}$ is the previous action. $g(\cdot)$ is a 2-layer FNN with 16 hidden units and a $sigmoid$ output layer.

We analyzed the attention matrix in the AttentionNeuron layer and visualized the attended positions. To be concrete, in CarRacing, the Query matrix has 1024 rows. Because we have $16 \times 16 = 256$ patches, the Key matrix has 256 rows, we therefore have an attention matrix of size $1024 \times 256$. To plot attended patches, we select from each row in the attention matrix the patch that has the largest value after softmax, this gives us a vector of length 1024. This vector represents the patches each of the 1024 output channels has considered to be the most important. 1024 is larger than the total patch count, however there are duplications (i.e. multiple output channels have mostly focused on the same patches). The unique number turns out to be $10 \sim 20$ at each time step. We emphasize these patches on the observation images to create an animation.