# OpenReview forum: "The Sensory Neuron as a Transformer: Permutation-Invariant Neural Networks for Reinforcement Learning"
_NeurIPS.cc/2021/Conference — NeurIPS 2021 Spotlight_

### Official Review · Reviewer_TUZP · 2021-07-11

**Rating:** 8
**Confidence:** 4

**Summary:**

This work unites previous research on Transformers and concepts from cellular automata to offer a novel model for learning global policies from local arbitrarily ordered variable-length sensory inputs. The authors evaluate their model using four standard RL tasks and show that their model, compared to the state-of-art, is robust to perturbations and occlusions of the inputs and also is significantly more effective for out-of-distribution generalization

**Limitations And Societal Impact:**

The authors have adequately addressed the limitations and potential negative societal impact of their work.

**Main Review:**

This work provides an interesting and highly valuable method for building RL models robust to various types of noise and capable of out-of-distribution generalization, which is a highly important and widely pursued goal in RL. To attain this goal, the authors propose a novel framework based on Set Transformers (a Transformer-based method for permutation-invariant neural-network analysis of datasets) and also based on concepts of cellular automata. The authors rigorously examine the properties of their new framework using a diverse set of standard RL tasks. As a result, the authors show impressive capabilities of their model to perform robustly in the presence of large-scale occlusions, noise, and out-of-distribution versions of the tasks. With that said, the text could use numerous clarifications and a comprehensive methods section, which would make the paper accessible for the broader NeurIPS community. Overall, the novel model, sound methodology, thorough testing, and highly significant results presented in this paper make it a valuable contribution for NeurIPS.

Specific comments:

Although the paper seems to have all necessary details to be understood, these details appear to be scattered throughout the text (and throughout the referenced literature) which may require reading the paper 2-3 times to cover all the important details. To this end, I suggest making use of unlimited supplementary space to write comprehensive methods, which – alone – would be sufficient to reproduce the model and the experiments.

The paper may also benefit from adding the details of the framework to the main text. Notably, the Set Transformers used here are much less known than the original Transformer framework, so it may be unfair to expect the reader to know its details or to be motivated to read a separate paper on it. As a good example, the Set Transformers paper itself starts with a brief yet sufficient introduction of the Transformer framework and explains the design choices for the Set Transformer framework immediately upon their appearance. One of such design choices – not explained in the current paper – is that SensoryNeuron, just like Set Transformers, attends the input data (Keys, Values) to a (smaller) set of separately learned/handcrafted Query embeddings – unlike the usual Transformers attending the elements of input data to each other. It is important to explain that learnable/predefined Query embeddings do not depend on the input and decrease the algorithm’s complexity from quadratic to linear.

The mathematical notation here can also be simplified, as it appears to have multiple variables which are not used. For example, the matrices Wq, Wk, and Wv were originally introduced in Transformer to be used in multi-head attention, where the different versions of these matrices were applied to the same Queries, Keys, and Values. It appears that the framework presented in this submission does not use multi-head attention, so these matrices can be sacrificed for the clarity of how the main model is explained. The matrix Wk, in particular, can be described together with the function Fk as both of them are implemented using neural networks – stacked with each other and thus forming a single neural network. The same goes for Fv and Wv – both being implemented as the identity matrices and thus unnecessary. Decomposition of Q and Wq is justified as it decreases the overall number of parameters and because Q is not learnable – but this distinction does not need to be immediately introduced. Focusing rather on the roles of Q, K, and V would make the paper more accessible for the broad audience.

Finally, I found the neuroscience-oriented introduction to this paper questionable. While the paper presents interesting, significant, and valuable results regarding the robustness of the algorithm to various types of noise, taking advantage of additional data, and out-of-distribution generalization – these results are summarized only at the end of the paper. At the same time, a reader who starts reading the text from the beginning may falsely think that the purpose of the paper is to learn from/replicate a biological phenomenon. Moreover, the introduction – in light of the details of the proposed model – may leave a reader with the impression that the brain information processing is permutation invariant, like in zero-shot learning, whereas it’s not. The neuroscience papers cited in the introduction, such as reverse biking or learning to “see” using an electrode matrix, clearly mention the significant time tolls of acquiring such skills. Although the presented work may have future neuroscience implications, the line of the argument currently presented here does not make a solid connection to neuroscience and, as such, needs to be revised/shrunk. Instead, the paper can start from the useful properties of the model, as outlined in the conclusions. The same goes for the abstract. The reclaimed space can be used for a detailed description of the SensoryNeuron, which is a central part of this work.

I would like to highlight here that all my suggestions above do not devalue the significance and the validity of the results. They are intended to highlight that a thorough explanation of the design choices, a broader overview of the Transformer, and starting from the robustness/generalization properties will make the paper a more accessible read for the broad NeurIPS audience. Even in the current format, this work is publishable in NeurIPS.

Minor comments:

-Why is the pass-through function Fv needed at all? Could you please briefly comment on the intuition why Fv, normally being a learnable parameter in Transformers, does not require learning here?

-Could you please also compute the statistical significance for the RL results? Given the huge standard deviation in the CartPole task, the SensoryNeuron’s apparent slight deficiency may not be statistically significant, which will strengthen your claim.

-Can you please explain why you spatially arrange the messages Mt in the AtariPong task? Naively, it would seem that, as you permute your sensory neurons, the 2D arrangement of their messages will be also random and the convolution on top of these messages will learn the same as the feedforward connectivity?

**Time Spent Reviewing:**

9

---

> ### Author Response · Authors · 2021-08-10
> **Author response to reviewer TUZP (R4)**
>
> *Note: we have referred to reviewer number rather than 4-letter code in our responses.*
>
> |  Reviewer  | Code   |
> |---|---|
> |  R1  |  dfwx  |
> |  R2  |  rWCR |
> |  R3  | Bz2K  |
> |  R4  | TUZP  |
>
> ---
>
> Thanks, R4, for your time and effort spent on reviewing this work, and for your comments, encouragement, and suggestions.
>
> Based on R4’s comments, we will incorporate the following changes in the text:
>
> 1. **Revise an shorten links to Neuroscience / psychology literature**
>
> We will take out the neuroscience part from the related work section, and also revise the intro to explain that neuroscience and psychology experiments helped inspire the formulation of the PI property in RL. As R4 pointed out, our approach is not biologically inspired, but rather, our aim is to directly perform shuffled sensory tasks in a zero-shot setting, something that us mere humans cannot do without devoting a significant amount of time to re-adjust. For the upside-down glass, left-right bicycle experiments, blind sensory substitution experiments mentioned, it is the fact that we *are* able to eventually perform them, that had inspired us to see if we can also get RL agents to explicitly work on shuffled tasks really well, but not in the same way that we ourselves perform them. To avoid giving the wrong impression, we will shrink / reward the abstract and intro, as R4 suggested.
>
> 2. **Improving Methods section and adding background material**
>
> The clarity of the methods section (with regards to both the writing and notation) will be improved (R3 had similar concerns). We would also like to add a background section on attention, permutation invariance, set transformer, prior to the methods section, as R4 suggested, so that we do not need to assume that the reader is already familiar with those concepts, thereby being able to engage a broader audience.. But rather than putting the background material in the Appendix as R4 suggested, we think it may be a good idea to incorporate it in the main text, which we believe to be manageable if we utilize the extra allotted page upon publication. If there isn’t enough space, we will put the background material in the Appendix as R4 suggested, and refer to it in the main text.
>
> We are encouraged by R4’s words that our results for PI RL agents and investigations on practical benefits of PI in RL are highly significant and valuable contributions to the NeurIPS community, and we would like to also improve the format to make it more broadly accessible for the entire NeurIPS audience, and not just to be read by the sub-community of members involved in attention and RL, so hopefully more people in the community can also explore this interesting area.
>
> __Our answers to specific questions asked by R4:__
>
> *> Why is the pass-through function Fv needed at all? Could you please briefly comment on the intuition why Fv, normally being a learnable parameter in Transformers, does not require learning here?*
>
> We use Fv as a convenient place to group data preprocessing operations. In vision tasks, Fv flattens the patch data for matrix multiplication with Wv. In non-vision tasks, the raw observations are in a good shape, hence Fv is a pass-through function. We isolate learnable parameters from Fv and put them in Wv (this is to show the formation of attention explicitly as we mentioned in Sec. 3). Wv also helps reshape the output data for the downstream network. AttentionNeuron’s output is Mt = \sigma( QWq x transpose(KWk) ) x (VWv), which is a MxN tensor where M is the number of rows in Q and N is the number of columns in Wv. For non-vision tasks, we use MLP policy and N=1 suffices, since V’s shape is |o|x1 we hence set Wv to identity. For vision tasks, we treat Wv as a parameter matrix, and learn it during training.
>
> *> Could you please also compute the statistical significance for the RL results? Given the huge standard deviation in the CartPole task, the SensoryNeuron’s apparent slight deficiency may not be statistically significant, which will strengthen your claim.*
>
> The large variances come from the environment because the initial states were sampled from a wide range of possible settings, some of which can lead to instant failure. We tried to estimate more accurate mean scores by running for 1000 tests instead of 100 (hence the estimate of the mean is accurate to the order of $\frac{1}{\sqrt{1000}}$ compared to $\frac{1}{\sqrt{100}}$ for other experiments). To quantify this uncertainty, we also ran t-tests. The results give a p-value small enough (i.e. < 0.01) to suggest that the difference is significant. It is also expected that the baseline slightly outperforms our method, because AttentionNeuron needs to spend some time steps in each episode to identify the inputs, whereas the baseline does not.
>
> *> Can you please explain why you spatially arrange the messages Mt in the AtariPong task? Naively, it would seem that, as you permute your sensory neurons, the 2D arrangement of their messages will be also random and the convolution on top of these messages will learn the same as the feedforward connectivity?*
>
> We agree with the reviewer that it is likely “the convolution on top of these messages will learn the same as the feedforward connectivity”. We had such arrangements in the paper because we wanted to show that AttentionNeuron is a general layer and by only reshaping the output, it is compatible with many existing policy networks, e.g., MLP for the Cartpole and Ant, Convnet for Pong and AttentionAgent for CarRacing.

---

> > ### Comment · Reviewer_TUZP · 2021-08-15
> > **Re: authors' response**
> >
> > I would like to thank the authors for their detailed responses.
> >
> > In rebuttal, the authors have proposed a reasonable scheme of rewriting the text to address the existing issues and make their text accessible to the broader audience.
> >
> > The authors have also performed additional experiments to address the reviewers' specific comments. In particular, they have compared the performance of their models to the more sophisticated baselines, which has strengthened the results of the paper.
> >
> > Overall, I believe that the authors have addressed the reviews diligently and adequately.

---

### Official Review · Reviewer_Bz2K · 2021-07-16

**Rating:** 7
**Confidence:** 3

**Summary:**

This paper presents a sensory neuron architecture as a layer in a neural network trained for reinforcement learning and claims that this sensory neuron makes a network generally permutation-invariant. There is an extensive related-work section covering examples of sensory modality adaptation in neuroscience, cellular automata, meta-learning, and attention, setting up the presented approach which uses collective intelligence of only distributed modules with only local awareness, self-organization through attention, and adaptation of an existing policy to be permutation invariant. The paper then presents a mathematical formulation for the AttentionNeuron architecture, including its temporal memory, to be used as the first layer in an RL agent. The paper next tests its approach on four different tasks, some with multiple environments and concludes with some discussion of the results which appear to be robust to permutation of the input vector unlike the baseline feedforward network.

**Limitations And Societal Impact:**

The limitations receive some discussion, though I think the paper would benefit from more beyond resource usage. I also don’t think the lack of invariance to action permutation is as significant as a lack of invariance to observation permutation would be, though this could be a limit of my own knowledge. Ablation experiments may have been helpful to yield this section. The societal impact section is considerable and far-reaching, and I think it briefly hits on many important points. I want to clarify that I think the way this paper is written calls for exactly this type of societal impact section, so the authors have done a good job. That said, I personally prefer depth and engagement with societal implications rather than a broad laundry list, and if the paper had a narrower focus, Section 6 could focus on just the implications of permutation invariance (i.e. some of its main applications) and/or RL agents.

**Main Review:**

*Originality:* I think this paper’s originality comes from the fact that it is a novel combination and then application of well-known techniques and concepts. The paper notes that Transformer’s attention mechanism is known to be permutation invariant and applies it here to reinforcement learning, which is the primary novelty I see in this work. The paper suggests an angle of collective intelligence due to the AttentionNeuron architecture, where each of N sensory neurons sees one of each of N observation components. It seems that the transformation from input to sensory neuron layer is a generalized permutation matrix, and the paper’s point is that this can be any generalized permutation matrix as long as the magnitude of each column remains the same (I think; I may not have all the details totally right). The connection to collective intelligence seems limited to a high-level abstraction, but I do think it’s an elegant and unique approach to permutation invariance whether or not it would be considered collective.

*Quality:* I think the methods and results of this paper are quite well-conceived and -executed. Though the method seems complex at first and to some degree needlessly complex, the discussion of design choices such as temporal memory and reduced network size is helpful. The solution is actually quite thoughtful, and the issues are more of presentation than content (discussed below). The experiments are impressive - the approach is used successfully in four different testbeds, some with multiple environments. Though there are some caveats with each one, particularly Atari Pong, the core message of permutation invariance seems to persist through the results. In my opinion, the paper is really strengthened by forgoing complicated cuts of data and various statistics to just show performance on a large range of testbeds. The only improvement may be ablations or analysis to rule out confounds.

*Clarity:* This paper is somewhat difficult to comprehend. Section-specific critique:

The related work is quite ambitious. I think all but the attention section could be collapsed into a section about permutation invariance and *maybe* collective intelligence, though as I said in *Originality*, that connection seems vague. Many of these are only related loosely, and the authors say as much at the end of each paragraph. This paper presents an interesting layer architecture that achieves a clear and concrete goal, and related work about that goal and similar architectures is enough in my opinion. This related work section is confusing and to some degree feels overpromising despite a paper that doesn’t need to overpromise. This makes the paper difficult to understand.

In the methods section, the notation is difficult to follow, partially because it is mostly in paragraph form. Figure 2 is quite helpful, though the 1:1 nature of the observation components to sensory neurons could be visually clearer, but the rest of the text on page 4 is dense with words and notation combined. What makes it more frustrating is that it is notation, not derivation or theory, so it shouldn’t be difficult to follow; a diagram or creative visual aids for the central definitions might help.

Finally, the results are very hefty. That’s one of the biggest value-adds of this paper, but it’s also overwhelming because it’s presented as one long laundry list. I realize that each testbed is very different so it may be hard to have a claims-driven structure to this section, but it would be nice to see unifying stats or even unifying qualitative analysis earlier than Section 5.

*Significance:* I think the ambitious and extensive related work section causes this paper to feel like it’s overstating its significance; I don’t think neuroscience is really a related field at all, nor am I convinced that this is an example of collective intelligence, nor do I know why it would make a difference. It’s also hard to tell if permutation invariance itself is significant - I think the significance of the main contribution comes from the fact that it’s 1) an elegant architectural approach (which builds on Transformer), unlike existing permutation-invariance works that require specific input formats and 2) application to reinforcement learning. If the language focused on what this paper is really doing, as opposed to big ideas that sound kind of like what it is doing, it would come across better. Nonetheless, I do think that this result is significant. The applications listed in section 5 are a good taste of what this method can do, and I suspect there are many more, and more immediate, applications that could be explored from this work.

**Time Spent Reviewing:**

6

---

> ### Author Response · Authors · 2021-08-10
> **Author response to reviewer Bz2K (R3)**
>
> *Note: we have referred to reviewer number rather than 4-letter code in our responses.*
>
> |  Reviewer  | Code   |
> |---|---|
> |  R1  |  dfwx  |
> |  R2  |  rWCR |
> |  R3  | Bz2K  |
> |  R4  | TUZP  |
>
> ---
>
> Thank you, R3, for your comments and suggestions.
>
> The suggestions in the presentation from R3 are insightful and valued. Specifically, we will incorporate some of R3’s suggestions, and will make modifications to the writing:
>
> - Removal of the neuroscience subsection in the Related Works section, and briefly mention inspiration from neuroscience and psychology experiments in the intro. We agree with R3 that this will make the paper easier to understand, and also shorten it.
>
> - In the methods section, we will create a clear and simple table of notation (now that we have more space by shortening the related work section), to complement Figure 2 and the text. This could be similar to Supplementary Material’s Table 1 (see pdf). We will also improve the dense text to make the methods section more clear and concise to read.
>
> In addition to the writing, we have included results of additional model analysis and ablation studies into the text, which will specifically include:
>
> - A study of the attention patches for visual tasks. In CarRacing, we see that the patches with high attention scores tend to focus on the road edges and on the car in the original CarRacing task, but the degree in which the patches are still attending to the road for the zero-shot generalization tasks vary. This study could help one investigate the limitations of the generalization abilities of the model. (See response to R1 for more details)
>
> - A further T-SNE analysis on the permutation invariant (PI) representation vector m_t on the Shuffled Pong experiment demonstrates that the model has learned to cluster similar states that require similar actions in similar regions in the embedding space, suggesting that much of the ‘heavy lifting’ is done at the PI representation learning level, which can lead to limitations depending on the usefulness of the PI representation space learned.
>
> - Ablation experiments conducted to examine the importance of feeding in the previous action, and the importance of the sensory neuron unit vs the attention mechanism (See responses to R1 and R2), which ties in with R3’s comment on the importance of PI vs collective intelligence (CI) concepts that are synthesized to produce this work.
>
> We are really happy to see that R3 believes our work to be a novel combination of ideas from PI and CI applied to reinforcement learning, and that our results are significant, as the applications listed in Section 5 are a glimpse of what can be done with PI RL agents. We look forward to also seeing the NeurIPS community incorporate ideas from PI RL and also work on further investigating this fascinating area of RL in the future.

---

### Official Review · Reviewer_rWCR · 2021-07-16

**Rating:** 6
**Confidence:** 3

**Summary:**

This work investigates the problem of shuffled or dropped out sensory inputs for reinforcement learning agents. The authors propose AttentionNeuron, a permutation-invariant attention layer operating on neurons which independently process the input currently present at its location. Each neuron utilizes its past context and the past action to interpret the seen input, and the attention mechanism combines the outputs of each neuron to generate a globally-aware embedding to be used for a standard policy architecture. The authors show that (1) AttentionNeuron is robust to shuffled and noisy observations on CartPole, (2) performing behavior cloning with an AttentionNeuron architecture can yield a policy robust to shuffling on Ant, (3) AttentionNeuron is robust to patch-level shuffling and dropouts on Pong, (4) AttentionNeuron improves robustness to drastic background changes on CarRacing, and (5) AttentionNeuron is robust to shuffling within a single episode.

**Limitations And Societal Impact:**

Yes, the authors adequately addressed the limitations and potential negative societal impact

**Main Review:**

Pros
1. Studies novel and biologically inspired permutation invariance question in the context of RL
2. Experiments evaluate on several diverse environments (CartPole, Ant, Pong, CarRacing)
3. Writing is generally clear

Cons
1. It is not clear to me that permutation invariance is a practically significant or useful property
2. Minimal ablations/comparisons to other methods in the experiments section

At a high level, I think this work is promising: it investigated a little-studied problem, permutation invariance to sensory observations in RL, motivated by observations in neuroscience regarding malleability in human brains. While it is not clear to me why this is a useful property from an engineering perpsective (i.e. as to why this can't be mostly tackled by the sensor interface), I think the direction studying biologically plausible RL/models is exciting. Furthermore, I think additional practical benefits gained as a result of permutation invariance (such as better generalization) are particularly exciting/important.

My main concerns are about the lack of comparisons/ablations to other methods, as the authors propose a method with several components but only compare to a feedforward network. I don't necessarily believe that my choices below would outperform the proposed method, but I think they would strengthen the paper and better contextualize the presented numbers. To propose a non-exhaustive but simple set of choices:
1. Transformer/vision transformer: a transformer operating directly on the sensory inputs/image patches without positional embeddings is permutation invariant in the same sense as AttentionNeuron, but is a simpler approach (i.e. AttentionNeuron without the LSTM input cells). I think this ablation would make clear whether or not the recurrent layers are necessary, and if it is important that the bottom-level cells "orient" themselves,or if simply the upper-level attention layers are capable of making sense of the disordered information. I also think a comparison with positional embeddings would be useful, as vision transformers currently represent a state-of-the-art in image-based processing.
2. LSTM after image processing: similarly to my above comment, this would clarify if this recurrent reorganization should be done at a low-level or a high-level. I think this is also biologically interesting, and practically it would be less computationally expensive since there wouldn't need to be an LSTM at every input location
3. Other generalization methods: as I mentioned above, I think a particularly exciting experiment is Section 4.4/Table 4 because it promises that generalization could be improved via permutation invariance. However, the experiment only compares to a minimal baseline. I think comparing to other representation learning methods, such as bisimulation [1] or contrastive learning [2] (or others), would make this experiment more exciting.

Overall, I think the direction toward permutation-invariant representations for sensory observations is underexplored and novel in the RL setting. Also, I think the authors investigate a few different interesting properties of the method, though I think the baselines are weak. As a result of the aforementioned weaknesses, I think this work currently does not meet the bar for publication. I am open to changing my score if the above issues are addressed, particularly if the comparisons to the transformer and high-level LSTM are included.

=====

References

[1] Zhang, Amy et al 2020. "Learning Invariant Representations for Reinforcement Learning without Reconstruction."

[2] Srinivas, Aravind et al 2020. "CURL: Contrastive Unsupervised Representations for Reinforcement Learning."

=====

Updates

I am satisfied with the authors' responses and additional baselines, therefore I am increasing my score from 4 to 6.

**Time Spent Reviewing:**

4

---

> ### Author Response · Authors · 2021-08-10
> **Author response to reviewer rWCR (R2)**
>
> *Note: we have referred to reviewer number rather than 4-letter code in our responses.*
>
> |  Reviewer  | Code   |
> |---|---|
> |  R1  |  dfwx  |
> |  R2  |  rWCR |
> |  R3  | Bz2K  |
> |  R4  | TUZP  |
>
> ---
>
> Thanks, R2, for your review and comments.
>
> We are glad to hear that R2 believes that our work is interesting, and we also think that the direction towards permutation-invariant (PI) representations for sensory observations is an underexplored and novel area in RL setting. Along with R3 and R4, we’re also excited about additional practical beneficial properties (such as robustness to noise / generalization / occlusion-resistant / variable number of sensors for an RL agent) as a result of PI.
>
> As the main concern R2 has is the lack of comparisons / ablations in the work, we have looked at R2’s suggestions and have run additional experiments to address these concerns. Below is a summary of what we have done:
>
> 1. We have run an ablation where we feed the sensory inputs directly to the transformer (i.e. AttentionNeuron without the LSTM input cells), for the cart-pole swing-up task.
>
> 2. Modifying (1), we replace the policy network (after the AttentionNeuron layer) from a feedforward network to a “high level” LSTM network.
>
> 3. We have performed experiments to compare the generalization results to the NetRand [lee2020] baseline, and spent some effort to improve the baseline method to make it perform as well as possible for our generalization task.
>
> For (1) and (2), we performed the suggested ablations on the non-vision tasks (i.e. cart-pole swing-up for now, and can run on ant later on if requested), since they are the tasks that use LSTM sensory neuron networks. As discussed in the paper, the vision tasks do not use RNNs/LSTMs due to practical implementation and memory issues, and use FNNs.
>
> __The results of Ablations (1) and (2) are in the following table:__
>
> | Model in paper  |  (1)  No LSTM in AttentionNeuron | (2) High level LSTM  |
> |---|---|---|
> | 472 ± 426  | 280 ± 272  | 287 ± 305  |
>
> We present the results from ablation studies in the table, the significance of performance differences are evident from looking at the policies qualitatively, and verified by statistical tests. The first column shows the result from our proposed method (copied from Table 1 in the paper). In the second column, we replaced the LSTM(in=2, out=16) with Linear(in=2, out=16, activation=”tanh”) in the AttentionNeuron layer and left the other settings unchanged. Without temporal memory, AttentionNeuron is unable to infer the identity of the inputs and thus failed to keep its test performance. In the third column, we followed the reviewer’s suggestion and moved the LSTM layer from AttentionNeuron to the policy network (i.e. low-level LSTM => high-level LSTM). Again, this model failed to match the proposed method, suggesting that temporal memory in the policy network cannot help AttentionNeuron identity inputs and generate a useful representation $m_t$ for the downstream controller, even if the controller has been upgraded to an LSTM from an FNN.
>
> __Discussion on the role of positional-embedding specifically in our method:__
>
> We also wish to clarify the usage of positional embeddings in the paper. In particular, unlike traditional Transformer models, we do not use positional embeddings to explicitly label or identify the position of each patch, since that will break PI. In our scheme, each patch becomes one row in the Key and the Value matrices, and because we don’t augment patches with positional embeddings, there is no specific id info the agent can rely on to infer the order if the rows in these matrices are permuted (hence PI property is preserved).
>
> The part where we do use the pos-embeddings is in the Query matrix. Because the Query matrix is not affected by permuted inputs, the PI property is still preserved, so we can fill it with positional embeddings (i.e., positions=row indices). In Vision transformers, the Query matrix is created from patch data (not hard-coded) and is hence affected by permuted inputs. In principle the Query matrix can be learned but as discussed in the paper, we find using a pos-embedding scheme for hard-coding the Query matrix is efficient and is straightforward to implement.
>
> __Baseline comparison for generalization (3):__
>
> For (3), we agree that our current work is lacking an adequate comparison with other generalization methods. This is something we also considered ourselves during the review period. Our understanding is that [zhang2020]’s approach trains an agent on a set of numerous modified (or ‘distracted’) environments (rather than the original environment), to demonstrate generalization to another set of modified environments. The focus of [srinivas2020] is using contrastive learning to improve the sample efficiency of RL agents, rather than on achieving OOD generalization performance for RL tasks. In our task we simply want to train our agent on the single original environment only, and demonstrate generalization performance.
>
> We decided to focus on a baseline method called NetRand [lee2020], a simple technique to improve the generalization ability of deep RL agents by using a randomized convnet that randomly perturbs input observations. In this method, the ‘augmentations’ are basically done internally in the randomized convnet, so in principle it can also be used to train an agent only on the original CarRacing environment to measure generalization performance.
>
> We do wish to emphasize that in this work, we do not set out to propose a method explicitly to tackle (and claim SOTA performance on) OOD generalization for RL tasks; our goal is to train agents to tackle permutation-invariant RL environments, and we discover that generalization is one of the interesting and practical consequences of PI RL agents.
>
> *Table: CarRacing Generalization Baseline Comparison*
>
> | Method | Training Env | KOF | Mt. Fuji | Ukiyoe | DS |
> |---|---|---|---|---|---|
> | NetRand + CNN | 480 ± 144 | 20 ± 84 | 356 ± 159 | 533 ± 111 | -27 ± 34 |
> | NetRand + AttentionAgent | 885 ± 64 | -52 ± 14 | 709 ± 94 | 656 ± 131 | 122 ± 134 |
> | Ours | 801 ± 147 | 646 ± 189 | 503 ± 152 | 661 ± 140 | 171 ± 146 |
>
> In the first row of the table above, we trained the NetRand baseline method on the base CarRacing task. The CarRacing task proved a bit difficult for the baseline method, which achieved a mean score of ~ 480, but despite having a lower mean score, it demonstrably generalizes to a few of the modified environments to achieve scores in a similar ballpark.
>
> In order to achieve a meaningful baseline comparison, we also spent some time improving the NetRand baseline method so that it can get close to SOTA (~ 900) on the base CarRacing task. To do that, we used NetRand as an input layer to the AttentionAgent (hard-attention) policy network, and trained the combination end-to-end using ES, which is consistent with our proposed method for this task. The NetRand+AttentionAgent combination achieved a respectable mean score of 885, and as we can see in the 2nd row of the above table, this approach also generalizes to a few of the unseen modifications of the CarRacing environment.
>
> Compared to our PI agent, our score on the base CarRacing task is lower, but this is expected since our agent would require some amount of time steps to identify each of the inputs (which could be shuffled), while the baseline does not (NetRand+AttentionAgent would simply fail on the shuffled versions of CarRacing). Despite this, our method still compares favourably on the generalization performance.
>
> We do wish to reiterate that it is not our goal in this work to propose a state-of-the-art solution for RL generalization. While methods such as NetRand were explicitly designed to handle certain types of RL generalization tasks well (and we have confirmed here that its generalization properties do work as advertised, with a bit of work), our work’s direction is on handling permutation invariant environments, and we simply want to show that the generalization property is an interesting consequence of PI RL agents. In principle, PI can even be combined with other methods focused on generalization to engineer stronger agents for this purpose.
>
> We thank R2 again for the comments and suggestions. We have included the ablation studies and baseline comparison results in the paper. We gave it our best effort to address the 3 points, and also worked to provide a meaningful strong baseline comparison for the generalization task (which R2 thought to be particularly exciting/important), so we hope that R2 can consider improving the score to reflect the concerns addressed. If there are other issues, we are happy to discuss further.
>
> [lee2020] Lee et al., Network Randomization: A Simple Technique for Generalization in Deep Reinforcement Learning, ICLR 2020.

---

> > ### Comment · Reviewer_rWCR · 2021-08-11
> > **Response to rebuttal**
> >
> > Thanks for running the new baselines! I have now increased my score from 4 to 6.

---

> > > ### Author Response · Authors · 2021-08-11
> > > **Thanks for increasing the score**
> > >
> > > Thanks for revising your assessment of the work. We've included the baseline results in the revised draft.
> > >
> > > -- Authors

---

### Official Review · Reviewer_dfwx · 2021-07-18

**Rating:** 6
**Confidence:** 4

**Summary:**

This paper extends the idea of set transformer to motor control, so that an agent can perform well even when its sensory input channels are shuffled and partly presented.
The performance is verified in cart-pole, ant, pong, and car racing tasks by using evolutionary optimization and behavioral closing of pre-trained policies.


**Limitations And Societal Impact:**

These are well considered and documented.

**Main Review:**

Originality:
The present result is an extension of the set transformer (ICML 2019). Although the title includes "reinforcement learning", actual learning takes evolutionary optimization and supervised learning (behavioral cloning).

Quality:
I wish the role of the previous action input to the attention network is better discussed. My guess is that the network compares the changes in the observations and action taken to infers which channel carries the critical information for control. It would be interesting to analyze which channels actually gathered attention (e.g., the patches including the car and road boundary?) and how that is affected if the past action is not presented.

Clarity:
The math is clearly presented.
In equation (1), most RL folks imagine Q as action values and V as state value, so their meanings as Query and Value may be better explained earlier.

Significance:
Although the architecture is reported to be inspired from sensory neurons, it has only remote link to neuroscience.
Although not tested in this paper, permutation-invariant feature would be helpful in the context of multi-agent learning, where an agent need to work with a variable number of agents coming and going in random order.




**Time Spent Reviewing:**

1h

---

> ### Author Response · Authors · 2021-08-10
> **Author response to reviewer dfwx (R1)**
>
> *Note: we have referred to reviewer number rather than 4-letter code in our responses.*
>
> |  Reviewer  | Code   |
> |---|---|
> |  R1  |  dfwx  |
> |  R2  |  rWCR |
> |  R3  | Bz2K  |
> |  R4  | TUZP  |
>
> ---
>
> Thanks, R1, for your review. We address your comments below:
>
> *​​> Originality: The present result is an extension of the set transformer (ICML 2019). Although the title includes "reinforcement learning", actual learning takes evolutionary optimization and supervised learning (behavioral cloning).*
>
> Our work certainly builds on the permutation invariant (PI) property of self-attention. While Set Transformer (Lee et al. 2019) explored the PI property of Transformer for machine learning tasks, our work here specifically investigates PI policies in reinforcement learning environments. Along with R2, R3, and R4, we believe this is an underexplored and novel research direction.
>
> We believe the paper is more than an extension of the Set Transformer paper. As R3 and R4 also pointed out, our work is a novel combination of concepts from Transformer’s attention and self-organization / cellular automata, two fairly distinct subareas in the field. Set Transformer alone wouldn’t be sufficient to address the tasks we have proposed in the work.
>
> Regarding whether this is considered *reinforcement learning*: The goal of the paper is also to train PI networks that can solve PI variations of reinforcement learning environments. Here, we consider ES to be a direct policy search method in RL. It has been demonstrated [1, 2] that CMA-ES is computing a natural policy gradient for updating the agent’s policy, which we argue is compatible with the RL paradigm. The focus of our work is not on devising RL algorithms in the traditional sense, but rather on the design of agents that can solve PI RL environments.
>
> Due to the increased difficulty of training PI RL agents, we also discuss the use of behavioral cloning (BC) as a practical method for converting an existing, non-PI RL policy to one that is PI, since it is relatively straightforward for readers to be able to train existing well-known algorithms to obtain a non-PI policy. In the Ant experiment, we even discussed and compared both the RL (via ES policy search) approach with BC, allowing the reader to access the practicalities and tradeoffs for training PI policies on RL environments.
>
> *> Quality: I wish the role of the previous action input to the attention network is better discussed. My guess is that the network compares the changes in the observations and action taken to infers which channel carries the critical information for control. It would be interesting to analyze which channels actually gathered attention (e.g., the patches including the car and road boundary?) and how that is affected if the past action is not presented.*
>
> We agree that the role of the previous action should be discussed in more detail, as it is one of the key components in our solution that enabled PI to work.
>
> The combination of observations and previous actions helps each sensory neuron to infer the casual relationship between each input channel and the applied actions, which allows them to identify which input stream they are fed. This is important because the temporal stream of inputs alone may not be sufficient. For instance, in Cartpole two observations are sin($\theta$) and cos($\theta$) which only differ by a phase shift, making it difficult to for a sensory neuron to distinguish between one vs the other; In the Ant locomotion task, most observations are also joint angles and angular velocities of the legs which are numerically similarly bounded and change in similar patterns.
>
> __Ablation study of the effect of previous action:__
>
> Based on R1’s suggestion, to verify the importance of the previous action, we conducted ablation studies in the Cart-pole Swing-up task when the past action is not added to AttentionNeuron (modified model). The mean score we have from the original model is 472, whereas the modified model gave a mean score of 372. Qualitatively, while the policy without the previous action can swing up the pole, and cannot reliably balance it for many time-steps.
>
> __Attention Patch Analysis for CarRacing:__
>
> Based on R1’s suggestion, we have also analyzed the attention patches in the CarRacing task to confirm that, as R1 correctly pointed out, the patches are primarily scattered around the road and usually on the car (though not at all time steps). We have created an animation to visualize this, but we are not sure we can upload it to an anonymized link to share in the rebuttal (we checked with a PC and they mentioned that the NeurIPS policy is unclear and advised us against uploading anonymized links).
>
> The following are more technical details as to how we visualized the attention, and how the patch analysis was conducted (Optional read):
>
> In the trained CarRacing agent, the Query matrix has 1024 rows. Because we have 16x16=256 patches, the Key matrix has 256 rows, we therefore have an attention matrix of size 1024x256 (i.e. A=softmax{ Q x transpose(K) } ). To plot attended patches, we select from each row in the attention matrix the patch that has the largest value after softmax, this gives us a vector of length 1024. This vector represents the patches each of the 1024 output channels has considered to be the most important. 1024 is larger than the total patch count, however there are duplications (i.e. multiple output channels have mostly focused on the same patches). The unique number turns out to be 10-20 at each time step. We emphasize these patches on the observation images to create an animation.
>
> *> Significance: Although the architecture is reported to be inspired from sensory neurons, it has only remote link to neuroscience. Although not tested in this paper, permutation-invariant feature would be helpful in the context of multi-agent learning, where an agent need to work with a variable number of agents coming and going in random order.*
>
> We agree that the link to neuroscience is only remote (mainly providing an inspirational role), so we have decided to shorten the related work section and cut down on the neuroscience discussion. We also agree that the PI property is indeed very useful to multi-agent RL (variable number of agents going in random order) and this has been explored in a few recent works (we cited a few of them, in particular, [Ref 50] in our paper). PI tasks for single-agent RL, however, haven't really been explored, which is why we decided to embark on this particular direction (rather than multi-agent direction) in this work.
>
> We hope that our additional results of analyzing the attention patches in the PI agent, and also the discussion / additional ablation studies of removing the previous action address R1’s concerns, and can encourage R1 to increase their score assessment of this work. We have included these additional ablation results (and attention patch analysis) in the paper.
>
> [1] Akimoto et al., Bidirectional Relation between CMA Evolution Strategies and Natural Evolution Strategies (2010)
>
> [2] Glasmachers et al., Exponential natural evolution strategies (2010)

---

> > ### Comment · Reviewer_dfwx · 2021-08-18
> > **clarification**
> >
> > Thanks for your clarifications and additional works.
> > Now I understand that behavioral cloning is not mandatory but an option to accelerate learning.
> > It was not clear (maybe I missed?) how many training time or episodes were used for each implementation. It was also not clear when sensory inputs were permuted in training; every time step, some times within each episode, at the beginning of each episode, or after a batch of episodes.
> > The suggested link with cellular automata was also not clear to me.
> > With the proposed revisions, I would raise my rating to above threshold.

---

> > > ### Author Response · Authors · 2021-08-18
> > > **Thank you for your response and clarification.**
> > >
> > > Thank you for your response and clarification.
> > >
> > > > It was not clear (maybe I missed?) how many training time or episodes were used for each implementation.
> > >
> > > For each task, we trained our agent until its performance has converged. Due to the space restrictions, we have listed the training time in the supplementary material (Section A.2).
> > >
> > > > It was also not clear when sensory inputs were permuted in training; every time step, some times within each episode, at the beginning of each episode, or after a batch of episodes.
> > >
> > > We didn’t permute the agent’s observations at all during training. We only permuted the observations in tests to verify the PI property.
> > >
> > > For all the tests (except those in Table 5), we shuffled the observations once at the beginning of each episode, and fixed the order until the end of the episode.
> > >
> > > To create table 5, we shuffled the observations every t steps during each episode, to show the extent in which the policy still works when the observations are shuffled several times during an episode.
> > >
> > > > The suggested link with cellular automata was also not clear to me.
> > >
> > > We were inspired by the self-organization aspect of cellular systems from CA’s (in particular the recent work in Neural CA’s (Mordvintsev et al,, 2020)), where we use distributed modules with only local awareness for each sensory neuron. But unlike CA’s, the self-organization happens through the attention mechanism, rather than through their immediate neighbors. We will clarify the link in the related work section, as mentioned in our general response to all reviewers. For example, we have emphasized the link to self-organized systems, instead of CA's explicitly, in the related work section.
> > >
> > > -- Authors

---

> > > > ### Author Response · Authors · 2021-08-25
> > > > **Checking in regarding clarification**
> > > >
> > > > Dear Reviewer dfwx,
> > > >
> > > > Thanks again for your comments earlier.
> > > >
> > > > Can you kindly let us know if we have provided clarification in the previous response to your comment?
> > > >
> > > > Happy to further clarify and revise if needed.
> > > >
> > > > -- Authors

---

> > > > > ### Comment · Reviewer_dfwx · 2021-08-28
> > > > > **sample complexity, identical modules**
> > > > >
> > > > > >>It was not clear (maybe I missed?) how many training time or episodes were used for each implementation.
> > > > > >For each task, we trained our agent until its performance has converged. Due to the space restrictions, we have listed the training time in the supplementary material (Section A.2).
> > > > > What I asked was not the machine time, but the number of time steps or episodes, i.e. sample complexity. These are critical information that are necessary for evaluating the goodness of any algorithm, so rather than hiding them in Appendix, they should be reported in the main performance comparison tables.
> > > > >
> > > > > >We didn’t permute the agent’s observations at all during training
> > > > > Now I realized that all SensoryNeurons are identical copies, so that permutation invariance is achieved by design, not by learning. I thought k in f_k was an index, but that was not the case. This point may be better made explicit.
> > > > >
> > > > > This reminds me a similarity with the MOSAIC architecture
> > > > > Haruno M, Wolpert DM, Kawato M (2001). Mosaic model for sensorimotor learning and control. Neural Comput, 13, 2201-20. https://doi.org/10.1162/089976601750541778
> > > > > in which each module produces the prediction output and a "responsibility signal" based on the prediction error. In their case difference modules receive the same inputs, and in your case identical modules receive different inputs.

---

> > > > > > ### Author Response · Authors · 2021-08-28
> > > > > > **Re: sample complexity, identical modules**
> > > > > >
> > > > > > We thank the reviewer for the reply and for the clarification and pointers. Below is our response:
> > > > > >
> > > > > > *> What I asked was not the machine time, but the number of time steps or episodes, i.e. sample complexity. These are critical information that are necessary for evaluating the goodness of any algorithm, so rather than hiding them in Appendix, they should be reported in the main performance comparison tables.*
> > > > > >
> > > > > > The training costs are summarized in the following table. A maximum of 20K generations is specified in the training, but stopped early if the performance converged. Each generation has 256x16=4096 episode rollouts, where 256 is the population size and 16 is the rollout repetitions. The Pong permutation-invariant (PI) agents were trained using behavior cloning on a pre-trained PPO policy (which is not PI-capable), with 10M training steps. Note that these settings were not chosen with sample-efficiency in mind, but rather for learning a performant PI-capable policy using distributed computation within a reasonable wall-clock time budget.
> > > > > >
> > > > > > |   | CartPole  | AntBullet | CarRacing  | Pong |
> > > > > > |---|---|---|---|---|
> > > > > > | # Generations  |  14K | 12K  | 4K  | -  |
> > > > > >
> > > > > > We wish to reiterate that sample efficiency is not the focus of this work, and that we are not trying to propose a new RL algorithm, but rather investigate interesting properties (such as robustness and generalization) of agents that are permutation invariant, regardless of how efficient their training was. It is this consideration combined with the space restriction that led us to put the computational info in the Appendix.
> > > > > >
> > > > > > We thank the reviewer for the suggestion and will describe the training costs in the main text, since future work that may focus on issues regarding sample efficiency for PI agents can easily compare with our results.
> > > > > >
> > > > > > A thought we also have is that we have also demonstrated that it *is* possible to simply use SOTA RL algorithms to learn a vanilla, non-PI policy (such as Pong), and then behavioral clone an agent to produce a PI version of the policy. So the question is whether there is much use in RL algorithms that explicitly increase the sample efficiency in a PI setting. However, we do think that an interesting future direction is to formulate environments where behavioral cloning *will* fail in a PI setting, and that interactions with the environment (in a PI setting) is *required* to learn a PI policy. For instance, we have demonstrated in AntBullet that behavioral cloning method requires the cloned agent to have a much larger number of parameters compared to one trained with RL. This is where an investigation in sample-efficiency improvements in the RL algorithm explicitly in the PI setting may be beneficial.
> > > > > >
> > > > > > *> This reminds me a similarity with the MOSAIC architecture
> > > > > > Haruno M, Wolpert DM, Kawato M (2001). Mosaic model for sensorimotor learning and control. Neural Comput, 13, 2201-20. https://doi.org/10.1162/089976601750541778
> > > > > > in which each module produces the prediction output and a "responsibility signal" based on the prediction error. In their case difference modules receive the same inputs, and in your case identical modules receive different inputs.*
> > > > > >
> > > > > > We thank the reviewer for the pointer, we will include this very interesting paper in our related works section. MOSAIC’s prediction error based controller selection mechanism resembles the sensory neuron message aggregation mechanism in our work, however, as the reviewer noted, we don’t learn multiple modules for this purpose, nor do we formulate or learn from prediction errors explicitly to guide the selection (which is done with forward models in each module of MOSAIC trained to predict the next state).
> > > > > >
> > > > > > -- Authors

---

### Author Response · Authors · 2021-08-10
**General Response to All Reviewers**

*Note: we have referred to reviewer number rather than 4-letter code in our responses.*

|  Reviewer  | Code   |
|---|---|
|  R1  |  dfwx  |
|  R2  |  rWCR |
|  R3  | Bz2K  |
|  R4  | TUZP  |

---

We want to thank our reviewers for taking the time to provide us with valuable feedback for this paper. We first want to discuss and clarify the two main contributions of this work:

1. We propose permutation invariant (PI) versions of standard RL environments, and devise agents that can solve them, using techniques from attention and self-organization. Note that our focus is on devising the agent, rather than proposing an RL algorithm. As such, we use simple direct policy search methods such as ES to train the proposed agent. To increase the usefulness of the idea, we also show that behavioral cloning is another possible method to convert a non-PI policy (easily trained using off-the-shelf RL methods like PPO) to obtain a PI version of the policy.

2. We investigate useful properties of such PI RL agents, such as robustness and generalization, and discuss their applications. We do wish to emphasize that our goal is not to propose a SOTA method explicitly to tackle generalization problems for RL tasks (R2 listed a few, and we have subsequently run a baseline comparison for generalization, discussed later), but rather to train agents to tackle PI RL environments, and in our work, we discover that generalization is one of the interesting and practical consequences of PI RL agents.

Reviewers generally agree on the novelty, significance, and impact of our work. R2 mentioned that “the direction toward permutation-invariant representations for sensory observations is underexplored and novel in the RL setting,” R3 thought our “paper’s originality comes from the fact that it is a novel combination and then application of well-known techniques and concepts,” and R4 encouragingly praised our “novel model, sound methodology, thorough testing, and highly significant results presented in this paper make it a valuable contribution for NeurIPS.” R3 also mentioned that results in our paper is “significant,” and that the “applications listed in section 5 are a good taste of what this method can do, and I suspect there are many more, and more immediate, applications that could be explored from this work.”

Taking reviewer feedback and suggestions into consideration, we have run the following experiments and analysis, which will be included in the paper:

1. We have run an analysis to see where the patches are being attended to by the agent in Shuffled CarRacing. We show that, indeed, the most important patches are located around the edge of the roads, and occasionally on the car. We have prepared animated visualizations to demonstrate this on the base CarRacing and the modified generalization tasks, so we can use this to analyze success / failures in unseen environments that test generalization. See response to R1/R3 for more details and discussion.

2. We have conducted an ablation study where we remove the previous-action from being observed by the sensory neurons, to demonstrate its importance. We have also explained in greater detail the role of previous-action in the paper. See response to R1 for more details.

3. We have run ablation studies where we remove the LSTM in the sensory neurons and connect the input signals directly to attention. In a further ablation experiment in this setup, we have also further replaced the downstream FNN controller with an LSTM. See response to R2 for more details.

4. For the CarRacing generalization task, we conducted a baseline comparison to NetRand (Lee et al., ICLR, 2020), a simple but effective technique developed to perform similar generalization tasks. The baseline method as is could not obtain an adequate score on the base CarRacing task (which is not a trivial environment to solve), and we first worked on improving the baseline method so that it can get close to SOTA on the base CarRacing task, and use this improved baseline to compare with our generalization results. See response to R2 for more details.

5. Based on R3’s feedback, we conducted a t-sne analysis on the PI representation vector $m_t$ on the Shuffled Pong experiment to demonstrate that the model has learned to cluster similar states that require similar actions in similar regions in the embedding space.

We will also improve the writing and clarity of the paper:

1. Based on R1, R3, R4’s recommendation, we will remove the neuroscience section from the related work, and shorten the abstract / introduction section. While the previous work on sensory substitution and related psychology experiments inspired us to propose challenging PI RL tasks, the method presented is obviously not happens in humans (most of us are probably not capable of playing “Shuffled Pong” in a zero-shot learning setting!), so we agree with R4 that it needs to be rephrased to not give the wrong impression to the reader.

2. To improve the Methods section, we added extra discussion on the importance of making the previous-action available to the sensory neurons (R1). We will also add a background material section before methods (R4), and make the methodology more clear (less dense), with an accompanying table to complement the figure and writing (R3).

We hope that these improvements will address reviewer concerns, and that reviewers can consider adjusting their scores accordingly. Thank you all again for helping us improve this paper!

---

### Decision · Program_Chairs · 2021-09-27

**Decision:**

Accept (Spotlight)

**Comment:**

I thank the authors for their submission and active participation in the discussions. Reviewer dfwx has pledged that with the revisions by the authors, they would raise their score. Since reviewer dfwx has not done that yet and did not actively participate in the discussion with the other reviewers, I assume a higher rating than the one currently recorded. Taking this into account, it seems reviewers unanimously agree that this paper is worthy of publication. In particular, reviewers appreciated the diverse set of environments used for evaluation [rWCR,TUZP], that is well executed [Bz2K] and that it presents an original synergy between existing techniques [Bz2K,TUZP]. During rebuttal and discussions, reviewer rWCR's concerns regarding baselines were addressed and reviewers TUZP and Bz2K agreed that the paper is improved based on the author rebuttal. Thus, I recommend acceptance and encourage the authors to further improve the clarity of their paper based on the reviewer feedback.